# Stein Self-Repulsive Dynamics: Benefits from Past Samples

**Mao Ye** [*]
UT Austin
my21@cs.utexas.edu

**Tongzheng Ren** [*]
UT Austin
tongzheng@utexas.edu

**Qiang Liu**
UT Austin
lqiang@cs.utexas.edu

## Abstract

We propose a new Stein self-repulsive dynamics for obtaining diversified samples from intractable un-normalized distributions. Our idea is to introduce Stein variational gradient as a repulsive force to push the samples of Langevin dynamics away from the past trajectories. This simple idea allows us to significantly decrease the auto-correlation in Langevin dynamics and hence increase the effective sample size. Importantly, as we establish in our theoretical analysis, the asymptotic stationary distribution remains correct even with the addition of the repulsive force, thanks to the special properties of the Stein variational gradient. We perform extensive empirical studies of our new algorithm, showing that our method yields much higher sample efficiency and better uncertainty estimation than vanilla Langevin dynamics.

## 1 Introduction

Drawing samples from complex un-normalized distributions is one of the most basic problems in statistics and machine learning, with broad applications to enormous research fields that rely on probabilistic modeling. Over the past decades, large amounts of methods have been proposed for approximate sampling, including both Markov Chain Monte Carlo (MCMC) [e.g., Brooks et al., 2011] and variational inference [e.g., Wainwright et al., 2008, Blei et al., 2017].

MCMC works by simulating Markov chains whose stationary distributions match the distributions of interest. Despite nice asymptotic theoretical properties, MCMC is widely criticized for its slow convergence rate in practice. In difficult problems, the samples drawn from MCMC are often found to have high auto-correlation across time, meaning that the Markov chains explore very slowly in the configuration space. When this happens, the samples returned by MCMC only approximate a small local region, and under-estimate the probability of the regions un-explored by the chain.

Stein variational gradient descent (SVGD) [Liu and Wang, 2016] is a different type of approximate sampling methods designed to overcome the limitation of MCMC. Instead of drawing random samples sequentially, SVGD evolves a pre-defined number of particles (or sample points) in parallel with a special interacting particle system to match the distribution of interest by minimizing the KL divergence. In SVGD, the particles interact with each other to simultaneously move towards the high probability regions following the gradient direction, and also move away from each other due to a special repulsive force. As a result, SVGD allows us to obtain diversified samples that correctly represent the variation of the distribution of interest. SVGD has found applications in various challenging problems [e.g., Feng et al., 2017, Haarnoja et al., 2017, Pu et al., 2017, Liu et al., 2017a, Gong et al., 2019]. See Han and Liu [e.g., 2018], Chen et al. [e.g., 2018], Liu et al. [e.g., 2019], Wang et al. [e.g., 2019a] for examples of extensions.

However, one problem of SVGD is that it theoretically requires to run an infinite number of chains in parallel in order to approximate the target distribution asymptotically [Liu, 2017]. With a finite

---

[*]Equal Contribution

number of particles, the fixed point of SVGD does still provide a *prioritized, partial* approximation to the distribution in terms of the expectation of a special case of functions [Liu and Wang, 2018]. Nevertheless, it is still desirable to develop a variant of "single-chain SVGD", which only requires to run a single chain sequentially like MCMC to achieve the correct stationary distribution asymptotically in time, with no need to take the limit of infinite number of parallel particles.

In this work, we propose an example of *single-chain SVGD* by integrating the special repulsive mechanism of SVGD with gradient-based MCMC such as Langevin dynamics. Our idea is to use repulsive term of SVGD to enforce the samples in MCMC away from the past samples visited at previous iterations. Such a new *self-repulsive dynamics* allows us to decrease the auto-correlation in MCMC and hence increase the mixing rate, but still obtain the same stationary distribution thanks to the special property of the SVGD repulsive mechanism. We provide thorough theoretical analysis of our new method, establishing its asymptotic convergence to the target distribution. Such result is highly non-trivial, as our new self-repulsive dynamic is a non-linear high-order Markov process. Empirically, we evaluate our methods on an array of challenging sampling tasks, showing that our method yields much better uncertainty estimation and larger effective sample size.

## 2   Background: Langevin dynamics & SVGD

This section gives a brief introduction on Langevin dynamics [Rossky et al., 1978] and Stein Variational Gradient Descent (SVGD) [Liu and Wang, 2016], which we integrate together to develop our new self-repulsive dynamics for more efficient sampling.

**Langevin Dynamics**   Langevin dynamics is a basic gradient based MCMC method. For a given target distribution supported on $\mathbb{R}^d$ with density function $\rho^*(\boldsymbol{\theta}) \propto \exp(-V(\boldsymbol{\theta}))$, where $V : \mathbb{R}^d \mapsto \mathbb{R}$ is the potential function, the (Euler-discrerized) Langevin dynamics simulates a Markov chain with the following rule:

$$\boldsymbol{\theta}_{k+1} = \boldsymbol{\theta}_k - \eta \nabla V(\boldsymbol{\theta}_k) + \sqrt{2\eta} \boldsymbol{e}_k, \qquad \boldsymbol{e}_k \sim \mathcal{N}(0, \mathbf{I}),$$

where $k$ denotes the number of iterations, $\{\boldsymbol{e}_k\}_{k=1}^{\infty}$ are independent standard Gaussian noise, and $\eta$ is a step size parameter. It is well known that the limiting distribution of $\boldsymbol{\theta}_k$ when $k \to \infty$ approximates the target distribution when $\eta$ is sufficiently small.

Because the updates in Langevin dynamics are local and incremental, new points generated by the dynamics can be highly correlated to the past sample, in which case we need to run Langevin dynamics sufficiently long in order to obtain diverse samples.

**Stein Variatinal Gradient Descent (SVGD)**   Different from Langevin dynamics, SVGD iteratively evolves a pre-defined number of particles in parallel. Starting from an initial set of particles $\{\boldsymbol{\theta}_0^i\}_{i=1}^M$, SVGD updates the $M$ particles in parallel by

$$\boldsymbol{\theta}_{k+1}^i = \boldsymbol{\theta}_k^i + \eta \boldsymbol{g}(\boldsymbol{\theta}_k^i; \hat{\delta}_k^M), \quad \forall i = 1, \dots, M,$$

where $\boldsymbol{g}(\boldsymbol{\theta}_k^i; \hat{\delta}_k^M)$ is a velocity field that depends the empirical distribution of the current set of particles $\hat{\delta}_k^M := \frac{1}{M} \sum_{j=1}^M \delta_{\boldsymbol{\theta}_k^j}$ in the following way,

$$\boldsymbol{g}(\boldsymbol{\theta}_k^i; \hat{\delta}_k^M) = \mathbb{E}_{\boldsymbol{\theta} \sim \hat{\delta}_k^M} \Big[ \underbrace{-K(\boldsymbol{\theta}, \boldsymbol{\theta}_k^i) \nabla V(\boldsymbol{\theta})}_{\text{Confining Term}} + \underbrace{\nabla_{\boldsymbol{\theta}} K(\boldsymbol{\theta}, \boldsymbol{\theta}_k^i)}_{\text{Repulsive Term}} \Big]$$

Here $\delta_{\boldsymbol{\theta}}$ is the Dirac measure centered at $\boldsymbol{\theta}$, and hence $\mathbb{E}_{\boldsymbol{\theta} \sim \hat{\delta}_k^M}[\cdot]$ denotes averaging on the particles. The $K(\cdot, \cdot)$ is a positive definite kernel, such as the RBF kernel, that can be specified by users.

Note that $\boldsymbol{g}(\boldsymbol{\theta}_k^i; \hat{\delta}_k^M)$ consists of a confining term and repulsive term: the confining term pushes particles to move towards high density region, and the repulsive term prevents the particles from colliding with each other. It is the balance of these two terms that allows us to asymptotically approximate the target distribution $\rho^*(\boldsymbol{\theta}) \propto \exp(-V(\boldsymbol{\theta}))$ at the fixed point, when the number of particles goes to infinite. We refer the readers to Liu and Wang [2016], Liu [2017], Liu and Wang [2018] for thorough theoretical justifications of SVGD. But a quick, informal way to justify the SVGD update is through the *Stein's identity*, which shows that the velocity field $\boldsymbol{g}(\boldsymbol{\theta}; \rho)$ equals zero when $\rho$ equals the true distribution $\rho^*$, that is, $\forall \boldsymbol{\theta}' \in \mathbb{R}^d$,

$$\boldsymbol{g}(\boldsymbol{\theta}'; \rho^*) = \mathbb{E}_{\boldsymbol{\theta} \sim \rho^*} [-K(\boldsymbol{\theta}, \boldsymbol{\theta}') \nabla V(\boldsymbol{\theta}) + \nabla_{\boldsymbol{\theta}} K(\boldsymbol{\theta}, \boldsymbol{\theta}')] = 0. \tag{1}$$

This equation shows that, the target distributions forms a fixed point of the update, and SVGD would converge if the particle distribution $\hat{\delta}_k^M$ gives a close approximation to the target distribution $\rho^*$.

## 3 Stein Self-Repulsive Dynamics

In this work, we propose to integrate Langevin dynamics and SVGD to simultaneously decrease the auto-correlation of Langevin dynamics and eliminate the need for running parallel chains in SVGD. The idea is to use Stein repulsive force between the the current particle and the particles from previous iterations, hence forming a new self-avoiding dynamics with fast convergence speed.

Specifically, assume we run a single Markov chain like Langevin dynamics, where $\boldsymbol{\theta}_k$ denotes the sample at the $k$-th iteration. Denote by $\tilde{\delta}_k^M$ the empirical measure of $M$ samples from the past iterations:

$$\tilde{\delta}_k^M := \frac{1}{M} \sum_{j=1}^{M} \delta_{\boldsymbol{\theta}_{k-jc_\eta}}, \qquad c_\eta = c/\eta,$$

where $c_\eta$ is a thinning factor, which scales inversely with the step size $\eta$, introduced to slim the sequence of past samples. Compared with the $\hat{\delta}_k^M$ in SVGD, which is averaged over $M$ parallel particles, $\tilde{\delta}_k^M$ is averaged across time over $M$ past samples. Given this, our Stein self-repulsive dynamics updates the sample via

$$\boldsymbol{\theta}_{k+1} \leftarrow \boldsymbol{\theta}_k \ + \ \underbrace{(-\eta V(\boldsymbol{\theta}_k) + \sqrt{2\eta}\boldsymbol{e}_k)}_{\text{Langevin}} \ + \ \underbrace{\eta\alpha\boldsymbol{g}(\boldsymbol{\theta}_k; \tilde{\delta}_k^M)}_{\text{Stein Repulsive}}, \tag{2}$$

in which the particle is updated with the typical Langevin gradient, plus a Stein repulsive force against the particles from the previous iterations. Here $\alpha \geq 0$ is a parameter that controls the magnitude of the Stein repulsive term. In this way, the particles are pushed away from the past samples, and hence admits lower auto-correlation and faster convergence speed. Importantly, the addition of the repulsive force *does not impact* the asymptotic stationary distribution, thanks to Stein's identity in (1). This is because if the self-repulsive dynamics has converged to the target distribution $\rho^*$, such that $\theta_k \sim \rho^*$ for all $k$, the Stein self-repulsive term would equal to zero in expectation due to Stein's identity and hence does not introduce additional bias over Langevin dynamics. Rigorous theoretical analysis of this idea is developed in Section 4.

**Practical Algorithm**     Because $\tilde{\delta}_k^M$ is averaged across the past samples, it is necessary to introduce a burn-in phase with the repulsive dynamics. Therefore, our overall procedure works as follows,

$$\boldsymbol{\theta}_{k+1} = \begin{cases} \boldsymbol{\theta}_k - \eta\nabla V(\boldsymbol{\theta}_k) + \sqrt{2\eta}\boldsymbol{e}_k, & k < Mc_\eta, \\ \boldsymbol{\theta}_k + \eta\left[-\nabla V(\boldsymbol{\theta}_k) + \alpha\boldsymbol{g}(\boldsymbol{\theta}_k; \tilde{\delta}_k^M)\right] + \sqrt{2\eta}\boldsymbol{e}_k, & k \geq Mc_\eta. \end{cases} \tag{3}$$

It includes two phases. The first phase is the same as the Langevin dynamics which collects the initial $M$ samples used in the second phase while serves as a warm start. The repulsive gradient update is introduced in the second phase to encourage the dynamics to visit the under-explored density region. We call this particular instance of our algorithm Self-Repulsive Langevin dynamics (SRLD), self-repulsive variants of more general dynamics is discussed in Section 5.

**Remark**     Note that the first phase is introduced to collect the initial $M$ samples. However, it's not really necessary to generate the initial $M$ samples with Langevin dynamics. We can simply use some other initialization distribution and get $M$ initial samples from that distribution. In practice, we find using Langevin dynamics to collect the initial samples is natural and it can also be viewed as the burn-in phase before sampling, so we use (3) in all of the other experiments.

**Remark**     The general idea of introducing self-repulsive terms inside MCMC or other iterative algorithms is not new itself. For example, in molecular dynamics simulations, an algorithm called metadynamics [Laio and Parrinello, 2002] has been widely used, in which the particles are repelled away from the past samples in a way similar to our method, but with a typical repulsive function, such as $\sum_j D(\theta_k, \theta_{k-jc_\eta})$, where $D(\cdot, \cdot)$ can be any kind of dis-similarity. However, introducing an arbitrary repulsive force would alter the stationary distribution of the dynamics, introducing a harmful bias into the algorithm. Besides, the self-adjusting mechanism in Deng et al. [2020] can also be viewed as a repulsive force using the multiplier in gradient. The key highlight of our approach,

as reflected by our theoretical results in Section 4, is the unique property of the Stein repulsive term, that allows us to obtain the correct stationary distribution even with the addition of the repulsive force.

**Remark**    Recent works [Gallego and Insua, 2018, Zhang et al., 2018] also combine SVGD with Langevin dynamics, in which, however, the Langevin force is directly added to a set of particles that evolve in parallel with SVGD. Using our terminology, their system is

$$\boldsymbol{\theta}_{k+1}^i = \boldsymbol{\theta}_k^i + (-\eta V(\boldsymbol{\theta}_k^i) + \sqrt{2\eta}\boldsymbol{e}_k^i) + \eta\alpha\boldsymbol{g}(\boldsymbol{\theta}_k^i;\ \hat{\delta}_k^M), \quad \boldsymbol{e}_k \sim \mathcal{N}(0, \mathbf{I}), \quad \forall i = 1, \dots, M.$$

This is significantly different from our method on both motivation and practical algorithm. Their algorithm still requires to simulate $M$ parallel chains of particles like SVGD, and was proposed to obtain easier theoretical analysis than the deterministic dynamics of SVGD. Our work is instead motivated by the practical need of decreasing the auto-correlation in Langevin dynamics, and avoiding the need of running multiple chains in SVGD, and hence must be based on *self-repulsion* against past samples along a single chain.

**An Illustrative Example**    We give an illustrative example to show the key advantage of our self-repulsive dynamics. Assume that we want to sample from a bi-variate Gaussian distribution shown in Figure 1. Unlike standard settings, we assume that we have already obtained some initial samples (yellow dots in Figure 1) before running the dynamics. The initial samples are assumed to concentrate on the left part of the target distribution as shown in Figure 1. In this extreme case, since the left part of the distribution is over-explored by the initial samples, it is desirable to have the subsequent new samples to concentrate more on the un-explored part of the distribution. However, standard Langevin dynamics does not take this into account, and hence yielding a biased overall representation of the true distribution (left panel). With our self-repulsive dynamics, the new samples are forced to explore the un-explored region more frequently, allowing us to obtain a much more accurate approximation when combining the new and initial samples.

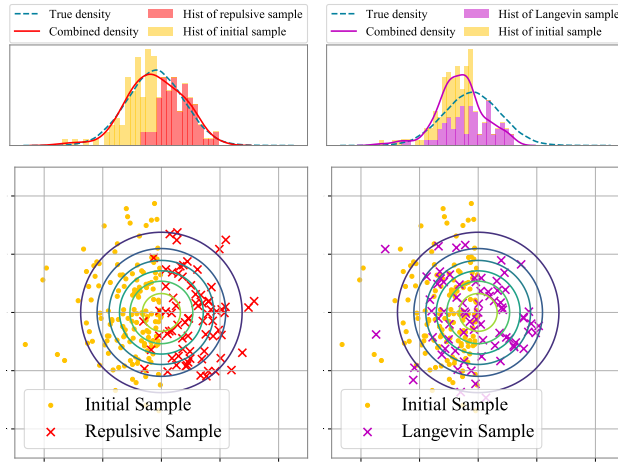

Figure 1: Illustrating the advantage of our Self-Repulsive Langevin dynamics. With a set of initial examples locating on the left part of the target distribution (yellow dots), Self-Repulsive Langevin dynamics is forced to explore the right part more frequently, yielding an accurate approximation when combined with the initial samples. Langevin dynamics, however, does not take the past samples into account and yields a poor overall approximation.

## 4    Theoretical Analysis of Stein Self-Repulsive Dynamics

We provide theoretical analysis of the self-repulsive dynamics. We establish that our self-repulsive dynamics converges to the correct target distribution asymptotically, in the limit when particle size $M$ approaches to infinite and the step size $\eta$ approaches to 0. This is a highly non-trivial task, as the self-repulsive dynamics is a highly complex, non-linear and high order Markov stochastic process. We attack this problem by breaking the proof into the following three steps:

(1) At the limit of $M \to \infty$ (called the **mean field limit**), we show that practical dynamics in (3) is closely approximated by a **discrete-time mean-field dynamics** characterized by (4).

(2) By further taking the limit of $\eta \to 0^+$ (called the **continuous-time limit**), the dynamics in (4) converges to a **continuous-time mean-field dynamics** characterized by (5).

(3) We show that the dynamics in (5) converges to the target distribution.

**Remark**    As we mentioned in Section 3, we introduce the first phase to collect the initial $M$ samples for the second phase, and our theoretical analysis follows this setting to make our theory as

close to the practice. However, the theoretical analysis can be generalized to the setting of drawing $M$ initial samples from some initialization distribution with almost identical argument.

**Notations** We use $\|\cdot\|$ and $\langle\cdot,\cdot\rangle$ to represent the $\ell_2$ vector norm and inner product, respectively. The Lipschitz norm and bounded Lipschitz norm of a function $f$ are defined by $\|f\|_{\mathrm{Lip}}$ and $\|f\|_{\mathrm{BL}}$. The KL divergence, Wasserstein-2 distance and Bounded Lipschitz distance between distribution $\rho_1$, $\rho_2$ are denoted as $\mathbb{D}_{\mathrm{KL}}[\rho_1\|\rho_2]$, $\mathbb{W}_2[\rho_1,\rho_2]$ and $\mathbb{D}_{\mathrm{BL}}[\rho_1,\rho_2]$, respectively.

## 4.1 Mean-Field and Continuous-Time Limits

To fix the notation, we denote by $\rho_k := \mathrm{Law}(\boldsymbol{\theta}_k)$ the distribution of $\boldsymbol{\theta}_k$ at time $k$ of the practical self-repulsive dynamics (3), which we refer to the **practical dynamics** in the sequel, when the initial particle $\boldsymbol{\theta}_0$ is drawn from an initial continuous distribution $\rho_0$. Note that given $\rho_0$, the subsequent $\rho_k$ can be recursively defined through dynamics (3). Due to the diffusion noise in Langevin dynamics, all $\rho_k$ are continuous distributions supported on $\mathbb{R}^d$. We now introduce the limit dynamics when we take the mean-field limit ($M \to +\infty$) and then the continuous-time limit ($\eta \to 0^+$).

**Discrete-Time Mean-Field Dynamics** ($M \to +\infty$) In the limit of $M \to \infty$, our practical dynamics (3) approaches to the following limit dynamics, in which the delta measures on the particles are replaced by the actual continuous distributions of the particles,

$$\tilde{\boldsymbol{\theta}}_{k+1} = \begin{cases} \tilde{\boldsymbol{\theta}}_k - \eta\nabla V(\tilde{\boldsymbol{\theta}}_k) + \sqrt{2\eta}\boldsymbol{e}_k, & k \le Mc_\eta, \\ \tilde{\boldsymbol{\theta}}_k + \eta\left[-\nabla V(\tilde{\boldsymbol{\theta}}_k) + \alpha\boldsymbol{g}(\tilde{\boldsymbol{\theta}}_k, \tilde{\rho}_k^M)\right] + \sqrt{2\eta}\boldsymbol{e}_k, & k \ge Mc_\eta. \end{cases} \tag{4}$$

where $\tilde{\rho}_k^M = \frac{1}{M}\sum_{j=1}^M \tilde{\rho}_{k-jc_\eta}$ and $\tilde{\rho}_k := \mathrm{Law}(\tilde{\boldsymbol{\theta}}_k)$ is the (smooth) distribution of $\tilde{\boldsymbol{\theta}}_k$ at time-step $k$ when the dynamics is initialized with $\tilde{\boldsymbol{\theta}}_0 \sim \tilde{\rho}_0 = \rho_0$. Compared with the practical dynamics in (3), the difference is that the empirical distribution $\tilde{\delta}_k^M$ is replaced by the smooth distribution $\tilde{\rho}_k^M$. Similar to the recursive definition of $\rho_k$ following dynamics (3), $\tilde{\rho}_k$ is also recursively defined through dynamics (4), starting from $\tilde{\rho}_0 = \rho_0$.

As we show in Theorem 4.3, if the auto-correlation of $\boldsymbol{\theta}_k$ decays fast enough and $M$ is sufficiently large, $\tilde{\rho}_k^M$ is well approximated by the empirical distribution $\tilde{\delta}_k^M$ in (3), and further the two dynamics ((3) and (4)) converges to each other in the sense that $\mathbb{W}_2[\rho_k, \tilde{\rho}_k] \to 0$ as $M \to \infty$ for any $k$. Note that in taking the limit of $M \to \infty$, we need to ensure that we run the dynamics for more than $Mc_\eta$ steps. Otherwise, SRLD degenerates to Langevin dynamics as we stop the chain before we finish collecting the $M$ samples.

**Continuous-Time Mean-Field Dynamics** ($\eta \to 0^+$) In the limit of zero step size ($\eta \to 0^+$), the discrete-time mean field dynamics in (4) can be shown to converge to the following continuous-time mean-field dynamics:

$$d\bar{\boldsymbol{\theta}}_t = \begin{cases} -\nabla V(\bar{\boldsymbol{\theta}}_t)dt + d\mathcal{B}_t, & t \in [0, Mc), \\ \left[-\nabla V(\bar{\boldsymbol{\theta}}_t) + \alpha\boldsymbol{g}(\bar{\boldsymbol{\theta}}_t, \bar{\rho}_t^M)\right]dt + d\mathcal{B}_t, & t \ge Mc. \end{cases} \tag{5}$$

where $\bar{\rho}_t^M := \frac{1}{M}\sum_{j=1}^M \bar{\rho}_{t-jc}(\cdot)$, $\mathcal{B}_t$ is the Brownian motion and $\bar{\rho}_t = \mathrm{Law}\left(\bar{\boldsymbol{\theta}}_t\right)$ is the distribution of $\bar{\boldsymbol{\theta}}_t$ at a continuous time point $t$ with $\boldsymbol{\theta}_0$ initialized by $\bar{\boldsymbol{\theta}}_0 \sim \tilde{\rho}_0 = \rho_0$. We prove that (5) is closely approximated by (4) with small step size in the sense that $\mathbb{D}_{\mathrm{KL}}[\tilde{\rho}_k\|\bar{\rho}_{k\eta}] \to 0$ as $\eta \to 0$ in Theorem 4.2, and importantly, the stationary distribution of (5) equals to the target distribution $\rho^*(\boldsymbol{\theta}) \propto \exp(-V(\boldsymbol{\theta}))$.

## 4.2 Assumptions

We first introduce the techinical assumptions used in our theoretical analysis.

**Assumption 4.1** (RBF Kernel). *We use RBF kernel, i.e. $K(\boldsymbol{\theta}_1, \boldsymbol{\theta}_2) = \exp(-\|\boldsymbol{\theta}_1 - \boldsymbol{\theta}_2\|^2/\sigma)$, for some fixed $0 < \sigma < \infty$.*

We only assume the RBF kernel for the simplicity of our analysis. However, it is straightforward to generalize our theoretical result to other positive definite kernels.

**Assumption 4.2** (V is dissipative and smooth). *Assume that $\langle\boldsymbol{\theta}, -\nabla V(\boldsymbol{\theta})\rangle \le b_1 - a_1\|\boldsymbol{\theta}\|^2$ and $\|\nabla V(\boldsymbol{\theta}_1) - \nabla V(\boldsymbol{\theta}_2)\| \le b_1\|\boldsymbol{\theta}_1 - \boldsymbol{\theta}_2\|$. We also assume that $\|\nabla V(\mathbf{0})\| \le b_1$. Here $a_1$ and $b_1$ are some finite positive constant.*

**Assumption 4.3** (Regularity Condition). *Assume* $\mathbb{E}_{\boldsymbol{\theta}\sim\rho_0}[\|\boldsymbol{\theta}\|^2] > 0$. *Define* $\rho_k^M = \sum_{j=1}^M \rho_{k-jc_\eta}/M$, *assume there exists* $a_2, B < \infty$ *such that*

$$\sup_{k \geq Mc_\eta} \frac{\mathbb{E}\left\|g(\boldsymbol{\theta}_k;\tilde{\delta}_k^M) - g(\boldsymbol{\theta}_k;\rho_k^M)\right\|^2}{\sup_{\|\boldsymbol{\theta}\|\leq B}\mathbb{E}\left\|g(\boldsymbol{\theta};\tilde{\delta}_k^M) - g(\boldsymbol{\theta};\rho_k^M)\right\|^2} \leq a_2.$$

**Assumption 4.4** (Strong-convexity). *Suppose that* $\langle \nabla V(\boldsymbol{\theta}_1) - \nabla V(\boldsymbol{\theta}_2), \boldsymbol{\theta}_1 - \boldsymbol{\theta}_2 \rangle \geq L\|\boldsymbol{\theta}_1 - \boldsymbol{\theta}_2\|^2$ *for a positive constant* $L$.

**Remark** Assumption 4.2 is standard in the existing Langevin dynamics analysis [see Dalalyan, 2017, Raginsky et al., 2017]. Assumption 4.3 is a weak condition as it assumes that the dynamics can not degenerate into one local mode and stop moving anymore. This assumption is expected to be true when we have diffusion terms like the Gaussian noises in our self-repulsive dynamics. Assumption 4.4 is a classical assumption on the existing Langevin dynamics analysis with convex potential Dalalyan [2017], Durmus et al. [2019]. Although being a bit strong, this assumption broadly applies to posterior inference problem in the limit of big data, as the posterior distribution converges to Gaussian distributions for large training set as shown by Bernstein-von Mises theorem. It is technically possible to further generalize our results to the non-convex settings with a refined analysis, which we leave as future work. This work focuses on the classic convex setting for simplicity.

### 4.3 Main Theorems

All of the proofs in this section can be found in Appendix E. We first prove that the limiting distribution of the continuous-time mean field dynamics (5) is the target distribution. This is achieved by writing dynamics (5) into the following (non-linear) partial differential equation:

$$\partial_t \bar{\rho}_t = \begin{cases} \nabla \cdot (-\nabla V \bar{\rho}_t) + \Delta \bar{\rho}_t & t \in [0, Mc) \\ \nabla \cdot \left[ \left(-\nabla V + \alpha g(\cdot, \bar{\rho}_t^M)\right) \bar{\rho}_t \right] + \Delta \bar{\rho}_t, & t \geq Mc. \end{cases}$$

**Theorem 4.1** (Stationary Distribution). *Given some finite* $M$, $c$ *and* $\alpha$, *and suppose that the limit distribution of* (5) *exists. Then the limit distribution is unique and satisfies* $\rho^*(\boldsymbol{\theta}) \propto \exp(-V(\boldsymbol{\theta}))$.

We then give the upper bound on the discretization error, which can be characterized by analyzing the KL divergence between $\tilde{\rho}_k$ and $\bar{\rho}_{k\eta}$.

**Theorem 4.2** (Time Discretization Error). *Given some sufficiently small step size* $\eta$ *and choose* $\alpha < a_2/(2b_1 + 4/\sigma)$. *Under Assumption 4.1, 4.2, 4.3 and* $c_\eta = c/\eta$. *we have for some constant* $C$,

$$\max_{l\in\{0,\ldots,k\}} \mathbb{D}_{\mathrm{KL}}\left[\bar{\rho}_{l\eta}\|\tilde{\rho}_l\right] \leq \begin{cases} \mathcal{O}\left(\eta + k\eta^2\right) & k \leq Mc_\eta - 1 \\ \mathcal{O}\left(\eta + Mc\eta + \alpha^2 Mce^{C\alpha^2(k\eta - Mc)}\eta^2\right) & k \geq Mc_\eta. \end{cases}$$

With this theorem, we can know that if $\eta$ is small enough, then the discretization error is small and $\tilde{\rho}$ approximates $\bar{\rho}$ closely. Next we give result on the mean field limit of $M \to \infty$.

**Theorem 4.3** (Mean-Field Limit). *Under Assumption 4.1, 4.2, 4.3, and 4.4, suppose that we choose* $\alpha$ *and* $\eta$ *such that* $-(a_1 - 2\alpha b_1/\sigma) + \eta b_1 < 0$; $\frac{2\alpha\eta}{\sigma}(b_1 + 1) < 1$; $a_2 - \alpha\left(2b_1 + \frac{4}{\sigma}\right) > 0$; *Then there exists a constant* $c_2$, *such that when* $L/a \geq c_2$ *and we have*

$$\mathbb{W}_2^2[\rho_k, \tilde{\rho}_k] = \begin{cases} \mathcal{O}\left(\alpha^2/M + \eta^2\right) & \geq Mc_\eta, \\ 0 & k \leq Mc_\eta - 1. \end{cases}$$

Thus, if $M$ is sufficiently large, $\rho_k$ can well approximate the $\tilde{\rho}_k$. Combining all the above theorems, we have the following Corollary showing the convergence of the proposed practical algorithm to the target distribution.

**Corollary 4.1** (Convergence to Target Distribution). *Under the assumptions of Theorem 4.1, 4.2 and 4.3, by choosing* $k, \eta, M$ *such that* $k\eta \to \infty$, $\exp(C\alpha^2 k\eta)\eta^2 = o(1)$ *and* $\frac{k\eta}{Mc} = \gamma(1 + o(1))$ *with* $\gamma > 1$, *we have*

$$\lim_{k,M\to\infty,\eta\to 0^+} \mathbb{D}_{BL}[\rho_k, \rho^*] = 0.$$

**Remark** A careful choice of parameters is needed as our system is a complicated longitudinal particle system. Also notice that if $\gamma \leq 1$, the repulsive dynamics reduces to Langevin dynamics, as only the samples from the first phase will be collected.

# 5 Extension to General Dynamics

Although we have focused on self-repulsive Langevin dynamics, our Stein self-repulsive idea can be broadly combined with general gradient-based MCMC. Following Ma et al. [2015], we consider the following general class of sampling dynamics for drawing samples from $p(\boldsymbol{\theta}) \propto \exp(-V(\boldsymbol{\theta}))$:

$$d\boldsymbol{\theta}_t = -\boldsymbol{f}(\boldsymbol{\theta})dt + \sqrt{2\boldsymbol{D}(\boldsymbol{\theta})}d\mathcal{B}_t,$$

$$\text{with} \quad \boldsymbol{f}(\boldsymbol{\theta}) = [\boldsymbol{D}(\boldsymbol{\theta}) + \boldsymbol{Q}(\boldsymbol{\theta})]\nabla V(\boldsymbol{\theta}) - \boldsymbol{\Gamma}(\boldsymbol{\theta}), \quad \Gamma_i(\boldsymbol{\theta}) = \sum_{j=1}^{d} \frac{\partial}{\partial \boldsymbol{\theta}_j} \left(D_{ij}(\boldsymbol{\theta}) + Q_{ij}(\boldsymbol{\theta})\right).$$

where $\boldsymbol{D}$ is a positive semi-definite diffusion matrix that determines the strength of the Brownian motion and $\boldsymbol{Q}$ is a skew-symmetric curl matrix that can represent the traverse effect [e.g. in Neal et al., 2011, Ding et al., 2014]. By adding the Stein repulsive force, we obtain the following general self-repulsive dynamics

$$d\bar{\boldsymbol{\theta}}_t = \begin{cases} -\boldsymbol{f}(\boldsymbol{\theta})dt + \sqrt{2\boldsymbol{D}(\boldsymbol{\theta})}d\mathcal{B}_t, & t \in [0, Mc) \\ -\left(\boldsymbol{f}(\boldsymbol{\theta}) + \alpha \boldsymbol{g}(\bar{\boldsymbol{\theta}}_t;\ \bar{\rho}_t^M)\right) dt + d\mathcal{B}_t, & t \geq Mc \end{cases} \tag{6}$$

where $\bar{\rho}_t := \text{Law}(\bar{\boldsymbol{\theta}}_t)$ is again the distribution of $\bar{\boldsymbol{\theta}}_t$ following (6) when initalized at $\bar{\boldsymbol{\theta}}_0 \sim \bar{\rho}_0$. Similar to the case of Langevin dynamics, this process also converges to the correct target distribution, and can be simulated by practical dynamics similar to (3).

**Theorem 5.1** (Stationary Distribution). *Given some finite $M$, $c$ and $\alpha$, and suppose that the limiting distribution of dynamics (6) exists. Then the limiting distribution is unique and equals the target distribution $\rho^*(\boldsymbol{\theta}) \propto \exp(-V(\boldsymbol{\theta}))$.*

# 6 Experiments

In this section, we evaluate the proposed method in various challenging tasks. We demonstrate the effectiveness of SRLD in high dimensions by applying it to sample the posterior of Bayesian Neural Networks. To demonstrate the superiority of the SRLD in obtaining diversified samples, we apply SRLD on contextual bandits problem, which requires the sampler efficiently explores the distribution in order to give good uncertainty estimation.

We include discussion on the parameter tuning and additional experiment on sampling high dimensional Gaussian and Gaussian mixture in Appendix B. Our code is available at `https://github.com/lushleaf/Stein-Repulsive-Dynamics`.

## 6.1 Synthetic Experiment

We first show how the repulsive gradient helps explore the whole distribution using a synthetic distribution that is easy to visualize. Following Ma et al. [2015], we compare the sampling efficiency on the following correlated 2D distribution with density

$$\rho^*([\theta_1, \theta_2]) \propto -\theta_1^4/10 - \left(4\left(\theta_2 + 1.2\right) - \theta_1^2\right)^2 /2.$$

We compare the SRLD with vanilla Langevin dynamics, and evaluate the sample quality by Maximum Mean Discrepancy (MMD) [Gretton et al., 2012], Wasserstein-1 Distance and effective sample size (ESS). Notice that the finite sample quality of gradient based MCMC method is highly related to the step size. Compared with Langevin dynamics, we have an extra repulsive gradient and thus we implicitly have larger step size. To rule out this effect, we set different step sizes of the two dynamics so that the gradient of the two dynamics has the same magnitude.

In addition, to decrease the influence of random noise, the two dynamics are set to have the same initialization and use the same sequence of Gaussian noise. We collect the sample of every iteration. We repeat the experiment 20 times with different initialization and sequence of Gaussian noise.

Figure 2 summarizes the result with different metrics. We can see that SRLD has a significantly smaller MMD and Wasserstein-1 Distance as well as a larger ESS compared with the vanilla Langevin dynamics. Moreover, the introduced repulsive gradient creates a negative auto-correlation between samples. Figure 3 shows a typical trajectory of the two sampling dynamics. We can see that

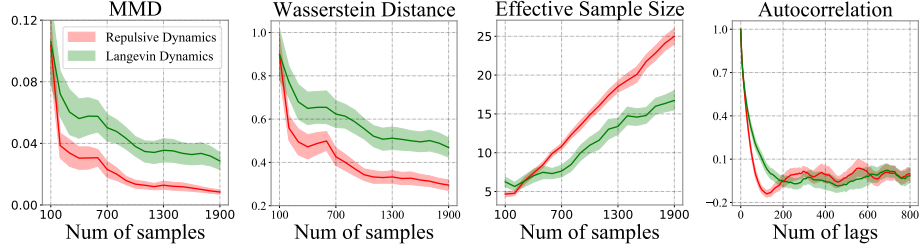

Figure 2: Sample quality of SRLD and Langevin dynamics for sampling the correlated 2D distribution. The auto-correlation is the averaged auto-correlation of the two dimensions.

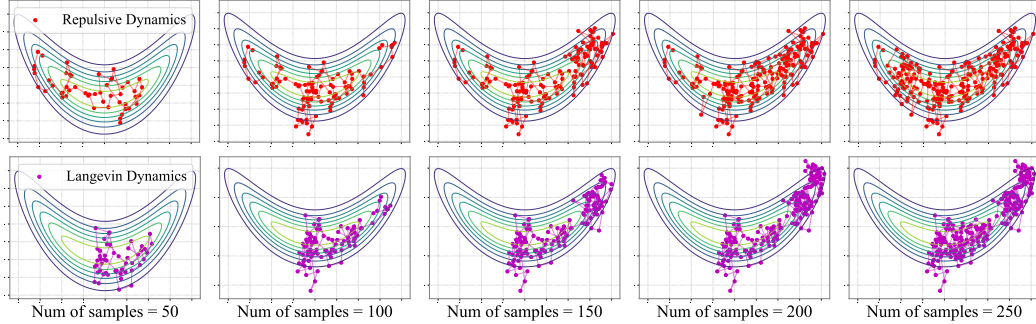

Figure 3: Sampling trajectory of the correlated 2D distribution.

SRLD have a faster mixing rate than vanilla Langevin dynamics. Note that since we use the same sequence of Gaussian noise for both algorithms, the difference is mainly due to the use of repulsive gradient rather than the randomness.

## 6.2 Bayesian Neural Network

Bayesian Neural Network is one of the most important methods in Bayesian Deep Learning with wide application in practice. Here we test the performance of SRLD on sampling the posterior of Bayesian Neural Network on the UCI datasets [Dua and Graff, 2017]. We assume the output is normal distributed, with a two-layer neural network with 50 hidden units and $\mathtt{tanh}$ activation to predict the mean of outputs. All of the datasets are randomly partitioned into $90\%$ for training and $10\%$ for testing. The results are averaged over 20 random trials. We refer readers to Appendix C for hyper-parameter tuning and other experiment details. Table 1 shows the average test RMSE and

| Dataset | Ave Test RMSE | | | Ave Test LL | | |
|---------|---------|---------|---------|---------|---------|---------|
| | SVGD | LD | SRLD | SVGD | LD | SRLD |
| Boston | $3.300 \pm 0.142$ | $3.342 \pm 0.187$ | $\mathbf{3.086 \pm 0.181}$ | $-4.276 \pm 0.217$ | $-2.678 \pm 0.092$ | $\mathbf{-2.500 \pm 0.054}$ |
| Concrete | $4.994 \pm 0.171$ | $4.908 \pm 0.113$ | $\mathbf{4.886 \pm 0.108}$ | $-5.500 \pm 0.398$ | $-3.055 \pm 0.035$ | $\mathbf{-3.034 \pm 0.031}$ |
| Energy | $0.428 \pm 0.016$ | $0.412 \pm 0.016$ | $\mathbf{0.395 \pm 0.016}$ | $-0.781 \pm 0.094$ | $-0.543 \pm 0.014$ | $\underline{\mathbf{-0.476 \pm 0.036}}$ |
| Naval | $0.006 \pm 0.000$ | $0.006 \pm 0.002$ | $\underline{\mathbf{0.003 \pm 0.000}}$ | $3.056 \pm 0.034$ | $4.041 \pm 0.030$ | $\mathbf{4.186 \pm 0.015}$ |
| WineRed | $0.655 \pm 0.008$ | $0.649 \pm 0.009$ | $\underline{\mathbf{0.639 \pm 0.009}}$ | $-1.040 \pm 0.018$ | $-1.004 \pm 0.019$ | $\underline{\mathbf{-0.970 \pm 0.016}}$ |
| WineWhite | $\mathbf{0.655 \pm 0.008}$ | $0.692 \pm 0.003$ | $0.688 \pm 0.003$ | $\underline{\mathbf{-1.040 \pm 0.019}}$ | $-1.047 \pm 0.004$ | $-1.043 \pm 0.004$ |
| Yacht | $0.593 \pm 0.071$ | $0.597 \pm 0.051$ | $\mathbf{0.578 \pm 0.054}$ | $-1.281 \pm 0.279$ | $-1.187 \pm 0.307$ | $\mathbf{-0.458 \pm 0.036}$ |

Table 1: Averaged test RMSE and test log-likelihood on UCI datasets. Results are averaged over 20 trials. The boldface indicates the method has the best average performance and the underline marks the methods that perform the best with a significance level of $0.05$.

test log-likelihood and their standard deviation. The method that has the best average performance is marked as boldface. We observe that a large portion of the variance is due to the random partition of the dataset. Therefore, to show the statistical significance, we use the matched pair $t$-test to test the statistical significance, mark the methods that perform the best with a significance level of 0.05 with underlines. Note that the results of SRLD/LD and SVGD is not very comparable, because SRLD/LD are single chain methods which averages across time, and SVGD is a multi-chain method that only use the results of the last iteration. We provide additional results in Appendix C that SRLD

| Dataset | SVGD | LD | SRLD |
|---|---|---|---|
| Mushroom | $20.7 \pm 2.0$ | $4.28 \pm 0.09$ | $\mathbf{3.80 \pm 0.16}$ |
| Wheel | $91.32 \pm 0.17$ | $38.07 \pm 1.11$ | $\mathbf{32.08 \pm 0.75}$ |

Table 2: Cumulative Regrets on two bandits problem (smaller is better). Results are averaged over 10 trails. Boldface indicates the methods with best performance and underline marks the best significant methods with significant level 0.05.

averaged on 20 particles (across time) can also achieve similar or better results as SVGD with 20 (parallel) particles.

### 6.3 Contextual Bandits

We consider the posterior sampling (a.k.a Thompson sampling) algorithm with Bayesian neural network as the function approximator, to demonstrate the uncertanty estimation provided by SRLD. We follow the experimental setting from Riquelme et al. [2018]. The only difference is that we change the optimization of the objective (e.g. evidence lower bound (ELBO) in variational inference methods) into running MCMC samplers. We compare the SRLD with the Langevin dynamics on two benchmarks from [Riquelme et al., 2018], and include SVGD as a baseline. For more detailed introduction, setup, hyper-parameter tuning and experiment details; see Appendix D.

The cumulative regret is shown in Table 2. SVGD is known to have the under-estimated uncertainty for Bayesian neural network if particle number is limited [Wang et al., 2019b], and as a result, has the worst performance among the three methods. SRLD is slightly better than vanilla Langevin dynamics on the simple Mushroom bandits. On the much more harder Wheel bandits, SRLD is significantly better than the vanilla Langevin dynamics, which shows the improving uncertainty estimation of our methods within finite number of samples.

## 7 Conclusion

We propose a Stein self-repulsive dynamics which applies Stein variational gradient to push samples from MCMC dynamics away from its past trajectories. This allows us to significantly decrease the auto-correlation of MCMC, increasing the sample efficiency for better estimation. The advantages of our method are extensive studied both theoretical and empirical analysis in our work. In future work, we plan to investigate the combination of our Stein self-repulsive idea with more general MCMC procedures, and explore broader applications.

**Broader Impact Statement** This work incorporates Stein repulsive force into Langevin dynamics to improve sample efficiency. It brings a positively improvement to the community such as reinforcement learning and Bayesian neural network that needs efficient sampler. Our work do not have any negative societal impacts that we can foresee in the future.

**Acknowledgement** This paper is supported in part by NSF CAREER 1846421, SenSE 2037267 and EAGER 2041327.

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
