[Supplementary Material]

# A Discussion on Hyper-parameter Tuning

The key hyper-parameters of SRLD are 1. $\alpha$, which balance the confining gradient and repulsive gradient; 2. $M$ the number of particles used; 3. $\sigma$ the bandwidth of kernel; 4. $\eta$ the stepsize; 5. $c_\eta$ the thinning factor. Among which, $\alpha$, $M$ and $\sigma$ are the hyper-parameter introduced by the proposed repulsive gradient and thus we mainly discuss these three hyper-parameter. The number of particles $M$ and bandwidth of kennel $\sigma$ are introduced by the use the repulsive term in SVGD [Liu and Wang, 2016]. In practice, we find using a similar setting for tuning $M$ and $\sigma$ as that in SVGD [Liu and Wang, 2016] gives good performance. In specific, in order to obtain good performance, $M$ does not needs be very large, and similar to SVGD, $M = 10$ already gives good enough particle approximation. A good choice of bandwidth $\sigma$ is also important for the kernel. In SVGD, instead of tuning $\sigma$, they propose an adaptive way to adjust $\sigma$ during the running of the dynamics. Specifically, they choose $\sigma = \mathrm{med}^2 / \log(M)$, where med is the median of the pairwise distance between the particles $\boldsymbol{\theta}_k^i$, $i \in [M]$. In this way, the bandwidth ensures that $\sum_{j=1}^{M} K(\boldsymbol{\theta}_k^i, \boldsymbol{\theta}_k^j) \approx 1$. This adaptive way of choosing $\sigma$ is also widely used in current approximation inference area, e.g., Liu et al. [2017b], Han and Liu [2017], Wang et al. [2019b,a]. We also find that applying this adaptive bandwidth is able to give good empirical performance and thus we also use this method in the implementation. Now we discuss how choose $\alpha$. Notice that $\alpha$ serves to balance the confining gradient and repulsive gradient and based on this motivation, we recommend readers to find a proper $\alpha$ using the samples at burn-in phase by setting

$$\alpha \approx \frac{\sum_{k=1}^{Mc_\eta} \|\nabla V(\boldsymbol{\theta}_k)\|}{\sum_{k=1}^{Mc_\eta} \left\| g(\boldsymbol{\theta}_k, \tilde{\delta}_k^M) \right\|}.$$

In this way, $\alpha$ balances the two kind of gradients. And then we may further tune $\alpha$ by searching around this value. An empirical good choice of $\alpha$ is 10 for the data sets we tested and we use $\alpha = 10$ for all the experiments.

The step size is important for gradient based MCMC, as too large step size gives too large discretization error while a too small step size will cause the dynamics converges very slowly. In this paper, we mainly use validation set to tune the step size. The thinning factor is also a common parameter is MCMC methods and usually MCMC methods are not sensitive to this parameter. SRLD is not sensitive to this parameter and we simply set $c_\eta = 100$ for all experiments.

# B Additional Experiment Result on Synthetic Data

In this section, we show additional experiment on synthetic data. To further visualize the role of the proposed stein repulsive gradient, we also apply our method to sample a 2D mixture of Gaussian distribution (see section B.1). To further study how different $\alpha$ influences sampling high dimension distribution, we apply SRLD to sample high dimensional Gaussian (section B.2) and high dimensional mixture of Gaussian (section B.3).

## B.1 Synthetic 2D Mixture of Gaussian Experiment

Figure 4: Sample quality and autocorrelation of the mixture distribution. The auto-correlation is the averaged auto-correlation of the two dimensions.

We aim to show how the repulsive gradient helps the particle escape from the local high density region by sampling the 2D mixture of Gaussian distribution using SRLD and Langevin dynamics.

Figure 5: Sampling trajectory of the mixture of Gaussian.

Figure 6: Sample quality and autocorrelation of the higher dimensional Gaussian distribution. The auto-correlation is the averaged auto-correlation of all dimensions.

The target density is set to be

$$\rho^*(\boldsymbol{\theta}) \propto 0.5 \exp\left(-\left\|\boldsymbol{\theta} - \mathbf{1}\right\|^2/2\right) + 0.5 \exp\left(-\left\|\boldsymbol{\theta} + \mathbf{1}\right\|^2/2\right),$$

where $\boldsymbol{\theta} = [\theta_1, \theta_2]^\top$ and $\mathbf{1} = [1, 1]^\top$. This target distribution have two mode at $-\mathbf{1}$ and $\mathbf{1}$, and vanilla Langevin dynamics can stuck in one mode while keeps the another mode under-explored (as the gradient of energy function can dominate the update of samples). We use the same evaluation method, step sizes, initialization and Gaussian noise as the previous experiment. We collect one sample every 100 iterations and the experiment is repeated for 20 times. Figure 4 shows that SRLD consistently outperforms the Langevin dynamics on all of the evaluation metrics.

To provide more evidence on the effectiveness of SRLD on escaping from local high density region, we plot the sampling trajectory of SRLD and vanilla Langevin dynamics on the mixture of Gaussian mentioned in Section 6.1. We can find that, when both of the methods obtain 200 samples, SRLD have started to explore the second mode, while vanilla Langevin dynamics still stuck in the original mode. When both of the methods have 250 examples, the vanilla Langevin dynamics just start to explore the second mode, while our SRLD have already obtained several samples from the second mode, which shows our methods effectiveness on escaping the local mode.

## B.2 Synthetic higher dimensional Gaussian Experiment

To show the performance of SRLD in higher dimensional case with different value of $\alpha$, we addition-ally considering the problem on sampling from Gaussian distribution with $d = 100$ and covariance $\boldsymbol{\Sigma} = 0.5\mathbf{I}$. We run SRLD with $\alpha = 100, 50, 20, 10, 0$ and the case $\alpha = 0$ reduces to Langevin. We collect 1 sample every 10 iterations. The other experiment setting is the same as the toy examples in the main text. The results are summarized at Figure 6. In this experiment, we set one SRLD with an inappropriate $\alpha = 100$. For this chain, the repulsive gradient gives strong repulsive force and thus has the largest ESS and the fastest decay of autocorrelation. While the inappropriate value $\alpha$ induces too much extra approximation error and thus its performance is not as good as these with smaller $\alpha$ (see MMD and Wasserstein distance). This phenomenon matches our theoretical finding.

Figure 7: Sample quality and autocorrelation of the higher dimensional mixture distribution. The auto-correlation is the averaged auto-correlation of all dimensions.

## B.3 Synthetic higher dimensional Mixture of Gaussian Experiment

We also consider sampling from the mixture of Gaussian with $d = 20$. The target density is set to be

$$\rho^*(\boldsymbol{\theta}) \propto \frac{1}{2} \exp\left(-0.5 \left\|\boldsymbol{\theta} - \sqrt{2/d}\mathbf{1}\right\|^2\right) + \frac{1}{2} \exp\left(-0.5 \left\|\boldsymbol{\theta} + \sqrt{2/d}\mathbf{1}\right\|^2\right),$$

where $\boldsymbol{\theta} = [\theta_1, ..., \theta_{20}]^\top$ and $\mathbf{1} = [1, ..., 1]^\top$. And thus the mean of the two mixture component is with distance $2\sqrt{2}$. We run SRLD with $\alpha = 20, 10, 5, 0$ (and when $\alpha = 0$, it reduces to LD). The other experiment setting is the same as the low dimensional mixture Gaussian case. Figure [7] summarizes the result. As shown in the figure, when $\alpha$ becomes larger, the repulsive forces helps the sampler better explore the density region.

## C BNN on UCI Datasets: Experiment Settings and Additional Results

We first give detailed experiment settings. We set a $\Gamma(1, 0.1)$ prior for the inverse output variance. We set the mini-batch size to be 100. We run 50000 iterations for each methods, and for LD and SRLD, the first 40000 iteration is discarded as burn-in. We use a thinning factor of $c_\eta = c/\eta = 100$ and in total we collect 100 samples from the posterior distribution. For each dataset, we generate 3 extra data splits for tuning the step size for each method. the number of past samples $M$ to be 10. In all experiments, we use RBF kernel with bandwidth set by the median trick as suggested in Liu and Wang [2016]. We use $\alpha = 10$ for all the data sets. For SVGD, we use the original implementation with 20 particles by Liu and Wang [2016].

We show some additional experiment result on posterior inference on UCI datasets. As mentioned in Section 6.2, the comparison between SVGD and SRLD is not direct as SVGD is a multiple-chain method with fewer particles and SRLD is a single chain method with more samples. To show more detailed comparison, we compare the SVGD with SRLD using the first 20, 40, 60, 80 and 100 samples, denoted as SRLD-$n$ where $n$ is the number of samples used. Table 3 shows the result of averaged test RMSE and table 4 shows the result of averaged test loglikelihood. For SRLD with different number of samples, the value is set to be boldface if it has better average performance than SVGD. If it is statistical significant with significant level 0.05 using a matched pair t-test, we add an underline on it.

Figure 8 and 9 give some visualized result on the comparison with Langevin dynamics and SRLD. To rule out the variance of different splitting on the dataset, the errorbar is calculated based on the difference between RMSE of SRLD and RMSE of Langevin dynamcis in 20 repeats (And similarily for test log-likelihood). And we only applied the error bar on Langevin dynamics.

| Dataset | Ave Test RMSE | | | | | |
|---|---|---|---|---|---|---|
| | SRLD-20 | SRLD-40 | SRLD-60 | SRLD-80 | SRLD-100 | SVGD |
| Boston | $3.236 \pm 0.174$ | $3.173 \pm 0.176$ | $3.130 \pm 0.173$ | $3.101 \pm 0.179$ | $3.086 \pm 0.181$ | $3.300 \pm 0.142$ |
| Concrete | $4.959 \pm 0.109$ | $4.921 \pm 0.111$ | $4.906 \pm 0.109$ | $4.891 \pm 0.108$ | $4.886 \pm 0.108$ | $4.994 \pm 0.171$ |
| Energy | $0.422 \pm 0.016$ | $0.409 \pm 0.016$ | $0.405 \pm 0.016$ | $0.399 \pm 0.016$ | $0.395 \pm 0.016$ | $0.428 \pm 0.016$ |
| Naval | $0.005 \pm 0.001$ | $0.004 \pm 0.000$ | $0.003 \pm 0.000$ | $0.003 \pm 0.000$ | $0.003 \pm 0.000$ | $0.006 \pm 0.000$ |
| WineRed | $0.654 \pm 0.009$ | $0.647 \pm 0.009$ | $0.644 \pm 0.009$ | $0.641 \pm 0.009$ | $0.639 \pm 0.009$ | $0.655 \pm 0.008$ |
| WineWhite | $0.695 \pm 0.003$ | $0.692 \pm 0.003$ | $0.690 \pm 0.003$ | $0.689 \pm 0.002$ | $0.688 \pm 0.003$ | $0.655 \pm 0.008$ |
| Yacht | $0.616 \pm 0.055$ | $0.608 \pm 0.052$ | $0.597 \pm 0.051$ | $0.587 \pm 0.054$ | $0.578 \pm 0.054$ | $0.593 \pm 0.071$ |

Table 3: Comparing SRLD with different number of samples with SVGD on test RMSE. The results are computed over 20 trials. For SRLD, the value is set to be boldface if it has better average performance than SVGD. The value if with underline if it is significantly better than SVGD with significant level 0.05 using a matched pair t-test.

| Dataset | Ave Test LL | | | | | |
|---|---|---|---|---|---|---|
| | SRLD-20 | SRLD-40 | SRLD-60 | SRLD-80 | SRLD-100 | SVGD |
| Boston | $-2.642 \pm .088$ | $-2.582 \pm 0.084$ | $-2.527 \pm 0.612$ | $-2.516 \pm 0.062$ | $-2.500 \pm 0.054$ | $-4.276 \pm 0.217$ |
| Concrete | $-3.084 \pm 0.036$ | $-3.061 \pm 0.034$ | $-3.050 \pm 0.033$ | $-3.040 \pm 0.031$ | $-3.034 \pm 0.031$ | $-5.500 \pm 0.398$ |
| Energy | $-0.580 \pm 0.053$ | $-0.536 \pm 0.048$ | $-0.522 \pm 0.046$ | $-0.504 \pm 0.044$ | $-0.476 \pm 0.036$ | $-0.781 \pm 0.094$ |
| Naval | $4.033 \pm 0.230$ | $4.100 \pm 0.171$ | $4.140 \pm 0.015$ | $4.167 \pm 0.014$ | $4.186 \pm 0.015$ | $3.056 \pm 0.034$ |
| WineRed | $-1.008 \pm 0.019$ | $-0.990 \pm 0.017$ | $-0.982 \pm 0.016$ | $-0.974 \pm 0.016$ | $-0.970 \pm 0.016$ | $-1.040 \pm 0.018$ |
| WineWhite | $-1.053 \pm 0.004$ | $-1.049 \pm 0.004$ | $-1.047 \pm 0.004$ | $-1.044 \pm 0.004$ | $-1.043 \pm 0.004$ | $-1.040 \pm 0.019$ |
| Yacht | $-1.160 \pm 0.256$ | $-0.650 \pm 0.173$ | $-0.556 \pm 0.096$ | $-0.465 \pm 0.037$ | $-0.458 \pm 0.036$ | $-1.281 \pm 0.279$ |

Table 4: Comparing SRLD with different number of samples with SVGD on test log-likelihood. The results are computed over 20 trials. For SRLD, the value is set to be boldface if it has better average performance than SVGD. The value if with underline if it is significantly better than SVGD with significant level 0.05 using a matched pair t-test.

# D   Contextual Bandit: Experiment Settings and More Background

Contextual bandit is a class of online learning problems that can be viewed as a simple reinforcement learning problem without transition. For a completely understanding of contextual bandit problems, we refer the readers to the Chapter 4 of [Bubeck et al., 2012]. Here we include the main idea for completeness. In contextual bandit problems, the agent needs to find out the best action given some observed context (a.k.a the optimal policy in reinforcement learning). Formally, we define $\mathcal{S}$ as the context set and $K$ as the number of action. Then we can concretely describe the contextual bandit problems as follows: for each time-step $t = 1, 2, \cdots, N$, where $N$ is some pre-defined time horizon (and can be given to the agent), the environment provides a context $s_t \in \mathcal{S}$ to the agent, then the agent should choose one action $a_t \in \{1, 2, \cdots, K\}$ based on context $s_t$. The environment will return a (stochastic) reward $r(s_t, a_t)$ to the agent based on the context $s_t$ and the action $a_t$ that similar to the reinforcement learning setting. And notice that, the agent can adjust the strategy at each time-step, so that this kinds of problems are called "online" learning problem.

Solving the contextual bandit problems is equivalent to find some algorithms that can minimize the pseudo-regret [Bubeck et al., 2012], which is defined as:

$$\overline{R}_N^{\mathcal{S}} = \max_{\pi:\mathcal{S}\rightarrow\{1,2,\cdots,K\}} \mathbb{E}\left[\sum_{t=1}^{N} r(s_t, g(s_t)) - \sum_{t=1}^{N} r(s_t, a_t)\right]. \tag{7}$$

where $\pi$ denotes the deterministic mapping from the context set $\mathcal{S}$ to actions $\{1, 2, \cdots, K\}$ (readers can view $\pi$ as a deterministic policy in reinforcement learning). Intuitively, this pseudo-regret measures the difference of cumulative reward between the action sequence $a_t$ and the best action sequence $\pi(s_t)$. Thus, an algorithm that can minimize the pseudo-regret (7) can also find the best $\pi$.

Posterior sampling [a.k.a. Thompson sampling; Thompson, 1933] is one of the classical yet successful algorithms that can achieve the state-of-the-art performance in practice [Chapelle and Li, 2011]. It works by first placing an user-specified prior $\mu_{s,a}^0$ on the reward $r(s, a)$, and each turn make decision based on the posterior distribution and update it, i.e. update the posterior distribution $\mu_{s,a}^t$ with the observation $r(s_{t-1}, a_{t-1})$ at time $t - 1$ where $a_{t-1}$ is selected with the posterior distribution: each time, the action is selected with the following way:

$$a_t = \underset{a\in\{1,2,\cdots,K\}}{\arg\max} \ \hat{r}(s_t, a), \quad \hat{r}(s_t, a) \sim \mu_{s,a}^t.$$

Figure 8: Comparison between SRLD and Langevin dynamics on test RMSE. The results are computed based on 20 repeats. The error bar is calculated based on RMSE of SRLD - RMSE of Langevin dynamics in 20 repeats to rule out the variance of different data splitting

i.e., greedy select the action based on the sampled reward from the posterior, thus called "Posterior Sampling". Algorithm 1 summarizes the whole procedure of Posterior Sampling.

---

**Algorithm 1** Posterior sampling for contextual bandits

---

**Input:** Prior distribution $\mu_{s,a}^0$, time horizon $N$
**for** time $t = 1, 2, \cdots, N$ **do**
    observe a new context $s_t \in \mathcal{S}$,
    sample the reward of each action $\hat{r}(s_t, a) \sim \mu_{s,a}^t, a \in \{1, 2, \cdots, K\}$,
    select action $a_t = \arg\max_{a \in \{1,2,\cdots,K\}} \hat{r}(s_t, a)$ and get the reward $r(s_t, a_t)$,
    update the posterior $\mu_{s_t,a_t}^{t+1}$ with $r(s_t, a_t)$.
**end for**

---

Notice that all of the reinforcement learning problems face the *exploration-exploitation dilemma*, so as the contextual bandit problem. Posterior sampling trade off the exploration and exploitation with the uncertainty provided by the posterior distribution. So if the posterior uncertainty is not estimated properly, posterior sampling will perform poorly. To see this, if we over-estimate the uncertainty, we can explore too-much sub-optimal actions, while if we under-estimate the uncertainty, we can fail to find the optimal actions. Thus, it is a good benchmark for evaluating the uncertainty provided by different inference methods.

Though in principle all of the MCMC methods return the samples follow the true posterior if we can run infinite MCMC steps, in practice we can only obtain finite samples as we only have finite time to run the MCMC sampler. In this case, the auto-correlation issue can lead to the under-estimate the uncertainty, which will cause the failure on all of the reinforcement learning problems that need exploration.

Here, we test the uncertainty provided by vanilla Langevin dynamics and Self-repulsive Langevin dynamics on two of the benchmark contextual bandit problems suggested by [Riquelme et al., 2018], called *mushroom* and *wheel*. One can read [Riquelme et al., 2018] to find the detail introduction of this two contextual bandit problems. For completeness, we include it as follows:

**Mushroom** Mushroom bandit utilizes the data from Mushroom dataset [Schlimmer, 1981], which includes different kinds of poisonous mushroom and safe mushroom with 22 attributes that can

Figure 9: Comparison between SRLD and Langevin dynamics on test log-likelihood. The results are computed based on 20 repeats. The error bar is calculated based on log-likelihood of SRLD - log-likelihood of Langevin dynamics in 20 repeats to rule out the variance of data splitting.

indicate whether the mushroom is poisonous or not. Blundell et al. [2015] first introduced the mushroom bandit by designing the following reward function: eating a safe mushroom will give a $+5$ reward, while eating a poisonous mushroom will return a reward $+5$ and $-35$ with equal chances. The agent can also choose not to eat the mushroom, which always yield a $0$ reward. Same to [Riquelme et al., 2018], we use $50000$ instances in this problem.

**Wheel** To highlight the need for exploration, [Riquelme et al., 2018] designs the wheel bandit, that can control the need of exploration with some "exploration parameter" $\delta \in (0, 1)$. The context set $\mathcal{S}$ is the unit circle $\|s\|_2 \leq 1$ in $\mathbb{R}^2$, and each turn the context $s_t$ is uniformly sampled from $\mathcal{S}$. $K = 5$ possible actions are provided: the first action yields a constant reward $r \sim \mathcal{N}(\mu_1, \sigma^2)$; the reward corresponding to other actions is determined by the provided context $s$:

- For $s \in \mathcal{S}$ s.t. $\|s\|_2 \leq \delta$, all of the four other actions return a suboptimal reward sampled from $\mathcal{N}(\mu_2, \sigma^2)$ for $\mu_2 < \mu_1$.

- For $s \in \mathcal{S}$ s.t. $\|s\|_2 > \delta$, according to the quarter the context $s$ is in, one of the four actions becomes optimal. This optimal action gives a reward of $\mathcal{N}(\mu_3, \sigma^2)$ for $\mu_3 \gg \mu_1$, and another three actions still yield the suboptimal reward $\mathcal{N}(\mu_2, \sigma^2)$.

Following the setting from [Riquelme et al., 2018], we set $\mu_1 = 1.2$, $\mu_2 = 1.0$, and $\mu_3 = 50$.

When $\delta$ approaches $1$, the inner circle $\|s\|_2 \leq \delta$ will dominate the unit circle and the first action becomes the optimal for most of the context. Thus, inference methods with poorly estimated uncertainty will continuously choose the suboptimal action $a_1$ for all of the contexts without exploration. This phenomenon have been confirmed in [Riquelme et al., 2018]. In our experiments, as we want to evaluate the quality of uncertainty provided by different methods, we set $\delta = 0.95$, which is pretty hard for existing inference methods as shown in [Riquelme et al., 2018], and use $50000$ contexts for evaluation.

**Experiment Setup** Following [Riquelme et al., 2018], we use a feed-forward network with two hidden layer of $100$ units and ReLU activation. We use the same step-size and thinning factor $c/\eta = 100$ for vanilla Langevin dynamics and SRLD, and set $M = 20$, $\alpha = 10$ on both of the mushroom and wheel bandits. The update schedule is similar to [Riquelme et al., 2018], and we just change the optimization step in stochastic variational inference methods into MCMC sampler step and replace the warm-up of stochastic variational inference methods in Riquelme et al. [2018] with

Figure 10: Visualization of the wheel bandit ($\delta = 0.95$), taken from [Riquelme et al., 2018].

the burn-in phase of the sampling. Similar to other methods in [Riquelme et al., 2018], we keep the initial learning rate as $10^{-1}$ for fast burn-in and the step-size for sampling is tuned on the mushroom bandit and keep the same for both the mushroom and wheel bandit. As this is an online posterior inference problem, we only use the last 20 samples to give the prediction. Notice that, in the original implementation of Riquelme et al. [2018], the authors only update a few steps with new observation after observing enough data, as the posterior will gradually converge to the true reward distribution and little update is needed after observing sufficient data. Similar to their implementation, after observing enough data, we only collect one new sample with the new observation each time. For SVGD, we use 20 particles to make the comparison fair, and also tune the step-size on the mushroom bandit.

# E    The Detailed analysis of SRLD

## E.1    Some additional notation

We use $\|\cdot\|_\infty$ to denote the $\ell_\infty$ vector norm and define the $\mathcal{L}_\infty$ norm of a function $f : \mathbb{R}^d \to \mathbb{R}^1$ as $\|f\|_{\mathcal{L}_\infty}$. $\mathbb{D}_{\mathrm{TV}}$ denote the Total Variation distance between distribution $\rho_1, \rho_2$ respectively. Also, as $K$ is $\mathbb{R}^d \times \mathbb{R}^d \to \mathbb{R}^1$, we denote $\|K\|_{\mathcal{L}_\infty,\mathcal{L}_\infty} = \sup_{\boldsymbol{\theta}_1,\boldsymbol{\theta}_2} K(\boldsymbol{\theta}_1,\boldsymbol{\theta}_2)$. For simplicity, we may use $\|K\|_{\infty,\infty}$ as $\|K\|_{\mathcal{L}_\infty,\mathcal{L}_\infty}$. In the appendix, we also use $\phi[\rho](\boldsymbol{\theta}) := \boldsymbol{g}(\boldsymbol{\theta};\rho)$, where $\boldsymbol{g}(\boldsymbol{\theta};\rho)$ is defined in the main text. For the clearance, we define $\pi_{M,c/\eta} * \rho_k := \rho_k^M$, $\pi_{M,c/\eta} * \tilde{\rho}_k := \tilde{\rho}_k^M$ and $\pi_{M,c} * \bar{\rho}_t := \bar{\rho}_t^M$, where $\rho_k^M$, $\tilde{\rho}_k^M$ and $\bar{\rho}_t^M$ are defined in main text.

## E.2    Geometric Ergodicity of SRLD

Before we start the proof of main theorems, we give the following theorem on the geometric ergodicity of SRLD. It is noticeable that under this assumption, the practical dynamics follows an $(Mc/\eta + 1)$-order nonlinear autoregressive model when $k \geq Mc/\eta$:

$$\boldsymbol{\theta}_{k+1} = \boldsymbol{\psi}\left(\boldsymbol{\theta}_k, ..., \boldsymbol{\theta}_{k-Mc/\eta}\right) + \sqrt{2\eta}\boldsymbol{e}_k,$$

where

$$\boldsymbol{\psi}\left(\boldsymbol{\theta}_k, ..., \boldsymbol{\theta}_{k-Mc/\eta}\right) = \boldsymbol{\theta}_k + \eta\left\{-\nabla V(\boldsymbol{\theta}_k) + \alpha\phi[\frac{1}{M}\sum_{j=1}^{M}\delta_{\boldsymbol{\theta}_{k-jc/\eta}}](\boldsymbol{\theta}_k)\right\}.$$

Further, if we stack the parameter by $\boldsymbol{\Theta}_k = \left[\boldsymbol{\theta}_k, ..., \boldsymbol{\theta}_{k-Mc/\eta}\right]^\top$ and define $\boldsymbol{\Psi}\left(\boldsymbol{\Theta}_k\right) = \left[\boldsymbol{\psi}^\top\left(\boldsymbol{\Theta}_k\right), \boldsymbol{\Theta}_k^\top\right]^\top$, we have

$$\boldsymbol{\Theta}_{k+1} = \boldsymbol{\Psi}\left(\boldsymbol{\Theta}_k\right) + \sqrt{2\eta}\boldsymbol{E}_k,$$

where $\boldsymbol{E}_k = \left[\boldsymbol{e}_k^\top, \boldsymbol{0}^\top, ..., \boldsymbol{0}^\top\right]^\top$. In this way, we formulate $\boldsymbol{\Theta}_k$ as a time homogeneous Markov Chain. In the following analysis, we only analyze the second phase of SRLD given some initial stacked particles $\boldsymbol{\Theta}_{Mc/\eta-1}$.

**Theorem E.1** (Geometric Ergodicity). *Under Assumption 4.1 and Assumption 4.2, suppose we choose $\eta$ and $\alpha$ such that*

$$\max\left(1 - 2\eta a_1 + \eta^2 b_1 + \frac{2\alpha\eta}{\sigma}b_1, \frac{2\alpha\eta}{\sigma}(b_1 + 1)\right) < 1,$$

*then the Markov Chain of $\boldsymbol{\Theta}_k$ is stationary, geometrically ergodic, i.e., for any $\boldsymbol{\Theta}_0' = \boldsymbol{\Theta}_{Mc/\eta-1}$, we have*

$$\mathbb{D}_{\mathrm{TV}}\left[P^k\left(\cdot, \boldsymbol{\Theta}_0\right), \Pi\left(\cdot\right)\right] \leq Q\left(\boldsymbol{\Theta}_0\right)e^{-rk},$$

*where $r = \mathcal{O}(\eta)$ is some positive constant, $Q(\boldsymbol{\Theta}_0)$ is constant related to $\boldsymbol{\Theta}_0$, $P^k$ is the $k$-step Markov transition kernel and $\Pi$ is the stationary distribution.*

We defer the proof to Appendix E.5.1.

## E.3    Moment Bound

**Theorem E.2** (Moment Bound). *Under Assumption 4.2, suppose that we have $\mathbb{E}_{\boldsymbol{\theta}\sim\rho_0}\|\boldsymbol{\theta}\|^2 < \infty$; and $a_2 - \alpha\|K\|_\infty\left(2b_1 + \frac{4}{\sigma}\right) > 0$, we have*

$$\sup_k \mathbb{E}_{\boldsymbol{\theta}\sim\rho_k}\|\boldsymbol{\theta}\|^2 \vee \sup_k \mathbb{E}_{\boldsymbol{\theta}\sim\tilde{\rho}_k}\|\boldsymbol{\theta}\|^2 \vee \sup_t \mathbb{E}_{\boldsymbol{\theta}\sim\bar{\rho}_t}\|\boldsymbol{\theta}\|^2$$

$$\leq \mathbb{E}_{\boldsymbol{\theta}\sim\rho_0}\|\boldsymbol{\theta}\|^2 + \frac{b_1 + 1 + \eta}{a_2 - \|K\|_{\mathcal{L}_\infty,\mathcal{L}_\infty}\frac{2\alpha}{\sigma} - \alpha\|K\|_{\mathcal{L}_\infty,\mathcal{L}_\infty}\left(2b_1 + \frac{2}{\sigma}\right)}.$$

*And by Lemma E.1, we thus have*

$$\sup_k \mathbb{E}_{\boldsymbol{\theta}\sim\rho_k}\|\nabla V(\boldsymbol{\theta})\|^2 \vee \sup_k \mathbb{E}_{\boldsymbol{\theta}\sim\tilde{\rho}_k}\|\nabla V(\boldsymbol{\theta})\|^2 \vee \sup_t \mathbb{E}_{\boldsymbol{\theta}\sim\bar{\rho}_t}\|\nabla V(\boldsymbol{\theta})\|^2$$

$$\leq b_1\mathbb{E}_{\boldsymbol{\theta}\sim\rho_0}\|\boldsymbol{\theta}\|^2 + \frac{b_1(b_1 + 1 + \eta)}{a_2 - \|K\|_{\mathcal{L}_\infty,\mathcal{L}_\infty}\frac{2\alpha}{\sigma} - \alpha\|K\|_{\mathcal{L}_\infty,\mathcal{L}_\infty}\left(2b_1 + \frac{2}{\sigma}\right)} + 1$$

The proof can be found at Appendix E.5.2.

### E.4 Technical Lemma

**Definition E.1** ($\alpha$-mixing)**.** *For any two $\sigma$-algebras $\mathcal{A}$ and $\mathcal{B}$, the $\alpha$-mixing coefficient is defined by*

$$\alpha(\mathcal{A}, \mathcal{B}) = \sup_{A \in \mathcal{A}, B \in \mathcal{B}} |\mathbb{P}(A \cap B) - \mathbb{P}(A)\mathbb{P}(B)|.$$

*Let $(X_k, k \geq 1)$ be a sequence of real random variable defined on $(\Omega, \mathcal{A}, \mathbb{P})$. This sequence is $\alpha$-mixing if*

$$\alpha(n) := \sup_{k \geq 1} \alpha(\mathcal{M}_k, \mathcal{G}_{k+n}) \to 0, \text{ as } n \to \infty,$$

*where $\mathcal{M}_j := \sigma(X_i, i \leq j)$ and $\mathcal{G}_j := \sigma(X_i, i \geq j)$ for $j \geq 1$. Alternatively, as shown by Theorem 4.4 of Bradley [2007]*

$$\alpha(n) := \frac{1}{4} \sup \left\{ \frac{\text{Cov}(f, g)}{\|f\|_{\mathcal{L}_\infty} \|g\|_{\mathcal{L}_\infty}}; \ f \in \mathcal{L}_\infty(\mathcal{M}_k), \ g \in \mathcal{L}_\infty(\mathcal{G}_{k+n}) \right\}.$$

**Definition E.2** ($\beta$-mixing)**.** *For any two $\sigma$-algebras $\mathcal{A}$ and $\mathcal{B}$, the $\alpha$-mixing coefficient is defined by*

$$\beta(\mathcal{A}, \mathcal{B}) := \sup \frac{1}{2} \sum_{i=1}^{I} \sum_{j=1}^{J} |\mathbb{P}(A_i \cap B_j) - \mathbb{P}(A_i)\mathbb{P}(B_j)|,$$

*where the supremum is taken over all pairs of finite partitions $\{A_1, ..., A_I\}$ and $\{B_1, ..., B_J\}$ of $\Omega$ such that $A_i \in \mathcal{A}$ and $B_j \in \mathcal{B}$ for each $i$, $j$. Let $(X_k, k \geq 1)$ be a sequence of real random variable defined on $(\Omega, \mathcal{A}, \mathbb{P})$. This sequence is $\beta$-mixing if*

$$\beta(n) := \sup_{k \geq 1} \beta(\mathcal{M}_k, \mathcal{G}_{k+n}) \to 0, \text{ as } n \to \infty.$$

**Proposition E.1** ($\beta$-mixing implies $\alpha$-mixing)**.** *For any two $\sigma$-algebras $\mathcal{A}$ and $\mathcal{B}$,*

$$\alpha(\mathcal{A}, \mathcal{B}) \leq \frac{1}{2}\beta(\mathcal{A}, \mathcal{B}).$$

This proposition can be found in Equation 1.11 of Bradley [2005].

**Proposition E.2.** *A (strictly) stationary Markov Chain is geometric ergodicity if and only if $\beta(n) \to 0$ at least exponentially fast as $n \to \infty$.*

This proposition is Theorem 3.7 of Bradley [2005].

**Lemma E.1** (Regularity Conditions)**.** *By Assumption 4.2, we have $\|\nabla V(\boldsymbol{\theta})\| \leq b_1 (\|\boldsymbol{\theta}_1\| + 1)$ and $\|\boldsymbol{\theta} - \eta \nabla V(\boldsymbol{\theta})\| \leq (1 - 2\eta a_1 + \eta^2 b_1) \|\boldsymbol{\theta}\|^2 + \eta^2 b_1 + 2\eta b_1$.*

**Lemma E.2** (Properties of RBF Kernel)**.** *For RBF kernel with bandwidth $\sigma$, we have $\|K\|_{\infty,\infty} \leq 1$ and*

$$\|K(\boldsymbol{\theta}', \boldsymbol{\theta}_1) - K(\boldsymbol{\theta}', \boldsymbol{\theta}_2)\| \leq \left\| e^{-(\cdot)^2/\sigma} \right\|_{\text{Lip}} \|\boldsymbol{\theta}_1 - \boldsymbol{\theta}_2\|_2$$

$$\|\nabla_{\boldsymbol{\theta}'} K(\boldsymbol{\theta}', \boldsymbol{\theta}_1) - \nabla_{\boldsymbol{\theta}'} K(\boldsymbol{\theta}', \boldsymbol{\theta}_2)\| \leq \left\| \frac{2}{\sigma} e^{-(\cdot)^2/\sigma}(\cdot) \right\|_{\text{Lip}} \|\boldsymbol{\theta}_1 - \boldsymbol{\theta}_2\|_2.$$

**Lemma E.3** (Properties of Stein Operator)**.** *For any distribution $\rho$ such that $\mathbb{E}_{\boldsymbol{\theta} \sim \rho} \|\nabla V(\boldsymbol{\theta})\| < \infty$, we have*

$$\|\phi[\rho](\cdot)\|_{\text{Lip}} \leq \left\| e^{-(\cdot)^2/\sigma} \right\|_{\text{Lip}} \mathbb{E}_{\boldsymbol{\theta} \sim \rho} \|\nabla V(\boldsymbol{\theta})\| + \left\| \frac{2}{\sigma} e^{-(\cdot)^2/\sigma}(\cdot) \right\|_{\text{Lip}},$$

$$\|\phi[\rho](\boldsymbol{\theta})\| \leq \|K\|_\infty \mathbb{E}_{\boldsymbol{\theta}' \sim \rho} \left[ \|\nabla V(\boldsymbol{\theta}')\| + \frac{2}{\sigma} (\|\boldsymbol{\theta}'\| + \|\boldsymbol{\theta}\|) \right]$$

$$\leq \|K\|_\infty b_1 + \mathbb{E}_{\boldsymbol{\theta}' \sim \rho} \left[ \left( \frac{2}{\sigma} + b_1 \right) \|\boldsymbol{\theta}'\| \right] + \|\boldsymbol{\theta}\|.$$

**Lemma E.4** (Bounded Lipschitz of Stein Operator). *Given $\boldsymbol{\theta}'$, define $\bar{\phi}_{\boldsymbol{\theta}'}(\boldsymbol{\theta}) := \phi[\delta_{\boldsymbol{\theta}'}](\boldsymbol{\theta}) = K(\boldsymbol{\theta}',\boldsymbol{\theta})\nabla V(\boldsymbol{\theta}') + \nabla_1 K(\boldsymbol{\theta}',\boldsymbol{\theta})$. We also denote $\bar{\phi}_{\boldsymbol{\theta}'}(\boldsymbol{\theta}) = [\bar{\phi}_{\boldsymbol{\theta}',1}(\boldsymbol{\theta}),...,\bar{\phi}_{\boldsymbol{\theta}',d}(\boldsymbol{\theta})]^\top$. We have*

$$\sum_{i=1}^{d} \left\| \bar{\phi}_{\boldsymbol{\theta}',i}(\boldsymbol{\theta}) \right\|_{\mathrm{Lip}}^2 \leq 2 \left\| \nabla V(\boldsymbol{\theta}') \right\|^2 \left\| e^{-\|\cdot\|^2/\sigma} \right\|_{\mathrm{Lip}}^2 + 2d \left\| \frac{2}{\sigma} e^{-\|\boldsymbol{\theta}\|^2/\sigma}\boldsymbol{\theta}_1 \right\|_{\mathrm{Lip}}^2$$

$$\sum_{i=d}^{d} \left\| \bar{\phi}_{\boldsymbol{\theta}',i}(\boldsymbol{\theta}) \right\|_{\mathcal{L}_\infty}^2 \leq 2d \left\| \frac{2}{\sigma} e^{-\|\boldsymbol{\theta}\|^2/\sigma}\boldsymbol{\theta}_1 \right\|_{\mathcal{L}_\infty}^2 + 2 \left\| e^{-\|\cdot\|^2/\sigma} \right\|_{\mathcal{L}_\infty}^2 \left\| \nabla V(\boldsymbol{\theta}') \right\|^2.$$

## E.5 Proof of Main Theorems

### E.5.1 Proof of Theorem E.1

The proof of this theorem is by verifying the condition of Theorem 3.2 of An and Huang [1996]. Suppose $\boldsymbol{\Theta} = [\boldsymbol{\theta}_1, ..., \boldsymbol{\theta}_{MC+1}]$, where $C = c/\eta$, we have

$$\|\boldsymbol{\psi}(\boldsymbol{\Theta})\| = \left\| \boldsymbol{\theta}_1 + \eta \left\{ -\nabla V(\boldsymbol{\theta}_1) + \alpha\phi[\frac{1}{M}\sum_{j=1}^{M}\delta_{\boldsymbol{\theta}_{1+jC}}](\boldsymbol{\theta}_k) \right\} \right\|$$

$$= \left\| \boldsymbol{\theta}_1 - \eta\nabla V(\boldsymbol{\theta}_1) + \frac{\eta\alpha}{M}\sum_{j=1}^{M}\left[ e^{-\|\boldsymbol{\theta}_{1+jC}-\boldsymbol{\theta}_1\|^2/\sigma}\frac{2}{\sigma}(\boldsymbol{\theta}_1-\boldsymbol{\theta}_{1+jC}) - e^{-\|\boldsymbol{\theta}_{1+jC}-\boldsymbol{\theta}_1\|^2/\sigma}\nabla V(\boldsymbol{\theta}_{1+jC}) \right] \right\|$$

$$\leq \left\| \boldsymbol{\theta}_1 - \eta\nabla V(\boldsymbol{\theta}_1) + \frac{2}{\sigma}\frac{\eta\alpha}{M}\sum_{j=1}^{M}e^{-\|\boldsymbol{\theta}_{1+jC}-\boldsymbol{\theta}_1\|^2/\sigma}\boldsymbol{\theta}_1 \right\|$$

$$+ \left\| \frac{\eta\alpha}{M}\sum_{j=1}^{M}e^{-\|\boldsymbol{\theta}_{1+jC}-\boldsymbol{\theta}_1\|^2/\sigma}\frac{2}{\sigma}(-\nabla V(\boldsymbol{\theta}_{1+jC})-\boldsymbol{\theta}_{1+jC}) \right\|$$

$$\leq \|\boldsymbol{\theta}_1 - \eta\nabla V(\boldsymbol{\theta}_1)\| + \frac{2\alpha\eta}{\sigma}\|K\|_{\infty,\infty}b_1(1+\|\boldsymbol{\theta}_1\|)$$

$$+ \frac{2\alpha\eta}{M\sigma}\sum_{j=1}^{M}\|K\|_{\infty,\infty}b_1\left(1+(1+\frac{1}{b_1})\|\boldsymbol{\theta}_{1+jC}\|\right)$$

$$\overset{(1)}{\leq} b_1(1+\frac{4\alpha\eta}{\sigma}\|K\|_{\infty,\infty}) + \eta^2 b_1 + 2\eta b_1$$

$$+ \left(1-2\eta a_1+\eta^2 b_1+\frac{2\alpha\eta}{\sigma}\|K\|_{\infty,\infty}b_1\right)\|\boldsymbol{\theta}_1\| + \frac{2\alpha\eta}{\sigma}\|K\|_{\infty,\infty}(b_1+1)\max_{i\in[MC+1]-\{1\}}\|\boldsymbol{\theta}_{1+jC}\|$$

$$\leq b_1(1+\frac{4\alpha\eta}{\sigma}\|K\|_{\infty,\infty}) + \eta^2 b_1 + 2\eta b_1$$

$$+ \max\left(1-2\eta a_1+\eta^2 b_1+\frac{2\alpha\eta}{\sigma}\|K\|_{\infty,\infty}b_1, \frac{2\alpha\eta}{\sigma}\|K\|_{\infty,\infty}(b_1+1)\right)\max_{i\in[MC+1]}\|\boldsymbol{\theta}_{1+jC}\|,$$

where (1) is by Lemma E.1. Thus, given the step size $\eta$, if we choose $\eta$, $\alpha$ such that

$$\max\left(1-2\eta a_1+\eta^2 b_1+\frac{2\alpha\eta}{\sigma}\|K\|_{\infty,\infty}b_1, \frac{2\alpha\eta}{\sigma}\|K\|_{\infty,\infty}(b_1+1)\right) < 1,$$

then our dynamics is geometric ergodic.

### E.5.2 Proof of Theorem E.2

**Continuous-Time Mean Field Dynamics** (5) Notice that as our dynamics has two phases and the first phase can be viewed as an special case of the second phase by setting $\alpha = 0$, here we only analysis the second phase. Define $U_t = \sup_{s\leq t}\sqrt{\mathbb{E}\left\|\bar{\boldsymbol{\theta}}_s\right\|^2}$, and thus

$$\frac{\partial}{\partial t}U_t^2 \leq \mathbb{E}\left\langle \bar{\boldsymbol{\theta}}_t, -V(\bar{\boldsymbol{\theta}}) + \alpha\phi[\pi_{M,c} * \bar{\rho}_t](\bar{\boldsymbol{\theta}}_t) \right\rangle \vee 0.$$

Now we bound $\mathbb{E}\left\langle \bar{\boldsymbol{\theta}}_t, -V(\bar{\boldsymbol{\theta}}) + \alpha\phi[\pi_{M,c} * \bar{\rho}_t](\bar{\boldsymbol{\theta}}_t) \right\rangle$:

$$
\begin{aligned}
&\mathbb{E}\left\langle \bar{\boldsymbol{\theta}}_t, -V(\bar{\boldsymbol{\theta}}_t) + \alpha\phi[\pi_{M,c} * \bar{\rho}_t](\bar{\boldsymbol{\theta}}_t) \right\rangle \\
&\leq b_1 - a_2 \mathbb{E}\left\| \bar{\boldsymbol{\theta}}_t \right\|^2 + \alpha \mathbb{E}\left\| \bar{\boldsymbol{\theta}}_t \right\| \left\| \phi[\pi_{M,c} * \bar{\rho}_t](\bar{\boldsymbol{\theta}}_t) \right\| \\
&\overset{(1)}{\leq} b_1 - a_2 \mathbb{E}\left\| \bar{\boldsymbol{\theta}}_t \right\|^2 + \alpha \left\| K \right\|_\infty \mathbb{E}\left\| \bar{\boldsymbol{\theta}}_t \right\| \mathbb{E}_{\boldsymbol{\theta}' \sim \pi_{M,c} * \bar{\rho}_t} \left[ \left\| \nabla V(\boldsymbol{\theta}') \right\| + \frac{2}{\sigma}\left( \|\boldsymbol{\theta}'\| + \|\boldsymbol{\theta}_t\| \right) \right] \\
&\leq b_1 - a_2 \mathbb{E}\left\| \bar{\boldsymbol{\theta}}_t \right\|^2 + \alpha \left\| K \right\|_\infty \mathbb{E}\left\| \bar{\boldsymbol{\theta}}_t \right\| \mathbb{E}_{\boldsymbol{\theta}' \sim \pi_{M,c} * \bar{\rho}_t} \left[ b_1\left( \|\boldsymbol{\theta}'\| + 1 \right) + \frac{2}{\sigma}\left( \|\boldsymbol{\theta}'\| + \|\boldsymbol{\theta}_t\| \right) \right] \\
&= b_1 - \left( a_2 - \left\| K \right\|_\infty \frac{2\alpha}{\sigma} \right) \mathbb{E}\left\| \bar{\boldsymbol{\theta}}_t \right\|^2 + \alpha \left\| K \right\|_\infty \mathbb{E}\left\| \bar{\boldsymbol{\theta}}_t \right\| \mathbb{E}_{\boldsymbol{\theta}' \sim \pi_{M,c} * \bar{\rho}_t} \left( \left( b_1 + \frac{2}{\sigma} \right) \|\boldsymbol{\theta}'\| + b_1 \right) \\
&\leq b_1 - \left( a_2 - \left\| K \right\|_\infty \frac{2\alpha}{\sigma} \right) U_t^2 + \alpha \left\| K \right\|_\infty \mathbb{E}\left\| \bar{\boldsymbol{\theta}}_t \right\| \mathbb{E}_{\boldsymbol{\theta}' \sim \pi_{M,c} * \bar{\rho}_t} \left( \left( b_1 + \frac{2}{\sigma} \right) \|\boldsymbol{\theta}'\| + b_1 \right) \\
&\leq b_1 - \left( a_2 - \left\| K \right\|_\infty \frac{2\alpha}{\sigma} \right) U_t^2 + \alpha \left\| K \right\|_\infty \left( b_1 + \frac{2}{\sigma} \right) \frac{1}{M}\sum_{j=1}^M U_t U_{t-jc} + \alpha \left\| K \right\|_\infty b_1 U_t \\
&\leq b_1 - \left( a_2 - \left\| K \right\|_\infty \frac{2\alpha}{\sigma} \right) U_t^2 + \alpha \left\| K \right\|_\infty \left( b_1 + \frac{2}{\sigma} \right) U_t^2 + \alpha \left\| K \right\|_\infty b_1 (U_t^2 + 1) \\
&\leq (b_1 + 1) - \left( a_2 - \left\| K \right\|_\infty \frac{2\alpha}{\sigma} - \alpha \left\| K \right\|_\infty \left( 2b_1 + \frac{2}{\sigma} \right) \right) U_t^2,
\end{aligned}
$$

where (1) is by E.3. By the assumption that $\lambda := a_2 - \left\| K \right\|_\infty \frac{2\alpha}{\sigma} - \alpha \left\| K \right\|_\infty \left( 2b_1 + \frac{2}{\sigma} \right) > 0$, we have

$$
\frac{\partial}{\partial t} U_t^2 \leq \left[ (b_1 + 1) - \lambda U_t^2 \right] \vee 0.
$$

By Gronwall's inequality, we have $U_t^2 \leq U_0^2 + \frac{b_1+1}{\lambda}$. (If $\frac{\partial}{\partial t} U_t^2 = 0$, then $U_t$ fix and this bound still holds.) Notice that in the first phase, as $\alpha = 0$, we have $\lambda < a_2$ and thus this inequality also holds.

**Discrete-Time Mean Field Dynamics** (4) Similarly to the analysis of the continuous-time mean field dynamics (5), we only give proof of the second phase. Define $U_k = \sup_{s \leq k} \sqrt{\mathbb{E}\left\| \tilde{\boldsymbol{\theta}}_s \right\|^2}$, and thus

$$
U_k^2 - U_{k-1}^2 \leq \left[ 2\eta \mathbb{E}\left\langle \tilde{\boldsymbol{\theta}}_{k-1}, -\nabla V(\tilde{\boldsymbol{\theta}}_k) + \alpha\phi[\pi_{M,c/\eta} * \tilde{\rho}_k](\tilde{\boldsymbol{\theta}}_k) \right\rangle + 2\eta^2 \right] \vee 0.
$$

By a similarly analysis, we have bound

$$
\begin{aligned}
&\mathbb{E}\left\langle \tilde{\boldsymbol{\theta}}_{k-1}, -\nabla V(\tilde{\boldsymbol{\theta}}_k) + \alpha\phi[\pi_{M,c/\eta} * \tilde{\rho}_k](\tilde{\boldsymbol{\theta}}_k) \right\rangle \\
&\leq (b_1 + 1) - \lambda U_t^2,
\end{aligned}
$$

where $\lambda = a_2 - \left\| K \right\|_{\infty,\infty} \frac{2\alpha}{\sigma} - \alpha \left\| K \right\|_{\infty,\infty} \left( 2b_1 + \frac{2}{\sigma} \right) > 0$. And thus we have

$$
U_k^2 - U_{k-1}^2 \leq \left[ 2\eta \left[ (b_1 + 1) - \lambda U_{k-1}^2 \right] + 2\eta^2 \right] \vee 0.
$$

It gives that

$$
U_k^2 \leq \frac{b_1 + 1 + \eta}{\lambda} + U_0^2.
$$

**Practical Dynamics** (3) The analysis of Practical Dynamics (3) is almost identical to that of the discrete-time mean field dynamics (4) and thus is omitted here.

### E.5.3 Proof of Theorem 4.1 and 5.1

Notice that the dynamics in Theorem 4.1 is special case of that in Theorem 5.1 and thus we only prove Theorem 5.1 here. After some algebra, we can show that the continuity equation of dynamics (6) is

$$
\partial_t \rho_t = \nabla \cdot \left( \left[ -\left( D(\boldsymbol{\theta}) + Q(\boldsymbol{\theta}) \right) \nabla V(\boldsymbol{\theta}) + \alpha\phi[\pi_{M,c} * \rho_t](\boldsymbol{\theta}_t) \right] \rho_t + \left( D(\boldsymbol{\theta}) + Q(\boldsymbol{\theta}) \right) \nabla \rho_t \right).
$$

Notice that the limiting distribution satisfies

$$
\begin{aligned}
0 \overset{a.e.}{=} & \nabla \cdot \left( \left[ -\left( D(\boldsymbol{\theta}) + Q(\boldsymbol{\theta}) \right) \nabla V(\boldsymbol{\theta}) + \alpha \phi [\pi_{M,c} * \rho_\infty](\boldsymbol{\theta}_t) \right] \rho_\infty + \left( D(\boldsymbol{\theta}) + Q(\boldsymbol{\theta}) \right) \nabla \rho_\infty \right) \\
= & \nabla \cdot \left( \left[ -\left( D(\boldsymbol{\theta}) + Q(\boldsymbol{\theta}) \right) \nabla V(\boldsymbol{\theta}) + \alpha \phi [\rho_\infty](\boldsymbol{\theta}_t) \right] \rho_\infty + \left( D(\boldsymbol{\theta}) + Q(\boldsymbol{\theta}) \right) \nabla \rho_\infty \right) \\
= & \nabla \cdot \left( \left[ -\left( D(\boldsymbol{\theta}) + Q(\boldsymbol{\theta}) \right) \nabla V(\boldsymbol{\theta}) \right] \rho_\infty + \left( D(\boldsymbol{\theta}) + Q(\boldsymbol{\theta}) \right) \nabla \rho_\infty \right) \\
& + \alpha \nabla \cdot \left( K * \left( \nabla \rho_\infty - \nabla V(\boldsymbol{\theta}) \rho_\infty \right) \rho_\infty \right).
\end{aligned}
$$

which implies that $\rho_\infty \propto \exp(-V(\boldsymbol{\theta}))$ is the stationary distribution.

### E.5.4 Proof of Theorem 4.2

In the later proof we use $c_d$ to represent the quantity

$$
\sqrt{ \mathbb{E}_{\boldsymbol{\theta} \sim \rho_0} \| \boldsymbol{\theta} \|^2 + \frac{b_1 + 1 + \eta}{a_2 - \|K\|_{\infty,\infty} \frac{2\alpha}{\sigma} - \alpha \|K\|_{\infty,\infty} \left( 2b_1 + \frac{2}{\sigma} \right)} }.
$$

Recall that there are two dynamics: the continuous-time mean field dynamics (5) and the discretized version discrete-time mean field Dynamics (4). Notice that here we couple the discrete-time mean field dynamics with the continuous-time mean field system using the same initialization. Given any $T = \eta N$, for any $0 \le t \le T$, define $\underline{t} = \lfloor \frac{t}{\eta} \rfloor \eta$. We introduce an another continuous-time interpolation dynamics:

$$
\begin{aligned}
\hat{\boldsymbol{\theta}}_t &= \begin{cases} -\nabla V(\hat{\boldsymbol{\theta}}_{\underline{t}}) + d\mathcal{B}_t, & t \in [0, Mc) \\ -\nabla V(\hat{\boldsymbol{\theta}}_{\underline{t}}) + \alpha \phi [\pi_{M,c} * \hat{\rho}_{\underline{t}}](\hat{\boldsymbol{\theta}}_{\underline{t}}) + d\mathcal{B}_t, & t \ge Mc, \end{cases} \\
\hat{\rho}_t &= \mathrm{Law}(\hat{\boldsymbol{\theta}}_t), \\
\hat{\boldsymbol{\theta}}_0 &= \bar{\boldsymbol{\theta}}_0 \sim \bar{\rho}_0,
\end{aligned}
$$

Notice that here we couples this interpolation dynamics with the same Brownian motion as that of the dynamics of $\bar{\boldsymbol{\theta}}_t$. By the definition of $\hat{\boldsymbol{\theta}}_t$, at any $t_k := k\eta$ for some integrate $k \in [N]$, $\hat{\boldsymbol{\theta}}_{t_k}$ and $\tilde{\boldsymbol{\theta}}_k$ has the same distribution. Define $\bar{\rho}_t^{\boldsymbol{\theta}_0} = \mathrm{Law}(\bar{\boldsymbol{\theta}}_t)$ conditioning on $\bar{\boldsymbol{\theta}}_0 = \boldsymbol{\theta}_0$ and $\hat{\rho}_t^{\boldsymbol{\theta}_0} = \mathrm{Law}(\hat{\boldsymbol{\theta}}_t)$ conditioning on $\hat{\boldsymbol{\theta}}_0 = \boldsymbol{\theta}_0$. Followed by the argument of proving Lemma 2 in Dalalyan [2017], if $k \ge \frac{Mc}{\eta}$, we have

$$
\begin{aligned}
& \mathbb{D}_{\mathrm{KL}} \left[ \bar{\rho}_{t_k}^{\boldsymbol{\theta}_0} \| \hat{\rho}_{t_k}^{\boldsymbol{\theta}_0} \right] \\
=& \frac{1}{4} \int_0^{t_k} \mathbb{E} \left\| -\nabla V(\hat{\boldsymbol{\theta}}_{\underline{s}}) + \alpha \phi [\pi_{M,c} * \hat{\rho}_{\underline{s}}](\hat{\boldsymbol{\theta}}_{\underline{s}}) + \nabla V(\hat{\boldsymbol{\theta}}_s) - \alpha \phi [\pi_{M,c} * \bar{\rho}_s](\hat{\boldsymbol{\theta}}_s) \right\|^2 ds \\
=& \frac{1}{4} \sum_{j=0}^{k-1} \int_{t_j}^{t_{j+1}} \mathbb{E} \left\| -\nabla V(\hat{\boldsymbol{\theta}}_{t_j}) + \alpha \phi [\pi_{M,c} * \hat{\rho}_{t_j}](\hat{\boldsymbol{\theta}}_{t_j}) + \nabla V(\hat{\boldsymbol{\theta}}_s) - \alpha \phi [\pi_{M,c} * \bar{\rho}_s](\hat{\boldsymbol{\theta}}_s) \right\|^2 ds \\
\le& \frac{3}{4} \sum_{j=0}^{k-1} \int_{t_j}^{t_{j+1}} \mathbb{E} \left\| \nabla V(\hat{\boldsymbol{\theta}}_{t_j}) - \nabla V(\hat{\boldsymbol{\theta}}_s) \right\|^2 ds \\
& + \frac{3\alpha^2}{4} \sum_{j=0}^{k-1} \int_{t_j}^{t_{j+1}} \mathbb{E} \left\| \phi [\pi_{M,c} * \hat{\rho}_{t_j}](\hat{\boldsymbol{\theta}}_{t_j}) - \phi [\pi_{M,c} * \bar{\rho}_s](\hat{\boldsymbol{\theta}}_{t_j}) \right\|^2 ds \\
\le& \frac{3\alpha^2}{4} \sum_{j=0}^{k-1} \int_{t_j}^{t_{j+1}} \mathbb{E} \left\| \phi [\pi_{M,c} * \bar{\rho}_s](\hat{\boldsymbol{\theta}}_{t_j}) - \phi [\pi_{M,c} * \bar{\rho}_s](\hat{\boldsymbol{\theta}}_s) \right\|^2 ds \\
=& I_1 + I_2 + I_3.
\end{aligned}
$$

We bound $I_1$, $I_2$ and $I_3$ separately.

**Bounding $I_1$ and $I_3$** By the smoothness of $\nabla V$, we have

$$
\left\| \nabla V(\hat{\boldsymbol{\theta}}_{t_j}) - \nabla V(\hat{\boldsymbol{\theta}}_s) \right\|^2 \le b_1^2 \left\| \hat{\boldsymbol{\theta}}_{t_j} - \hat{\boldsymbol{\theta}}_s \right\|^2.
$$

And by Lemma E.3 (Lipschitz of Stein Operator), we know that

$$\|\phi[\pi_{M,c} * \bar{\rho}_s](\boldsymbol{\theta}_1) - \phi[\pi_{M,c} * \bar{\rho}_s](\boldsymbol{\theta}_2)\|$$
$$\leq \left[\left\|e^{-(\cdot)^2/\sigma}\right\|_{\text{Lip}} \mathbb{E}_{\boldsymbol{\theta} \sim \pi_{M,c}*\bar{\rho}_s} \|\nabla V(\boldsymbol{\theta})\| + \left\|\frac{2}{\sigma}e^{-(\cdot)^2/\sigma}(\cdot)\right\|_{\text{Lip}}\right] \|\boldsymbol{\theta}_1 - \boldsymbol{\theta}_2\|_2 .$$

And by the Assumption 4.2 and that $\bar{\rho}_s$ as finite second moment, we have

$$\|\phi[\pi_{M,c} * \bar{\rho}_s](\boldsymbol{\theta}_1) - \phi[\pi_{M,c} * \bar{\rho}_s](\boldsymbol{\theta}_2)\|$$
$$\leq C c_d \|\boldsymbol{\theta}_1 - \boldsymbol{\theta}_2\|_2 .$$

Combine the two bounds, we have

$$I_1 + I_3 \leq \frac{3Cc_d^2}{4} \sum_{j=0}^{k-1} \int_{t_j}^{t_{j+1}} \mathbb{E} \left\|\hat{\boldsymbol{\theta}}_{t_j} - \hat{\boldsymbol{\theta}}_s\right\|^2 ds.$$

Notice that $\hat{\boldsymbol{\theta}}_t = \hat{\boldsymbol{\theta}}_{\underline{t}} + \left[-\nabla V(\hat{\boldsymbol{\theta}}_{\underline{t}}) + \alpha\phi[\pi_{M,c} * \hat{\rho}_{\underline{t}}](\hat{\boldsymbol{\theta}}_{\underline{t}})\right](t - \underline{t}) + \int_{\underline{t}}^{t} d\mathcal{B}_s$. By Itô's lemma, it implies that

$$I_1 + I_3 \leq \frac{3Cc_d^2}{4} \sum_{j=0}^{k-1} \int_{t_j}^{t_{j+1}} \mathbb{E} \left\|\hat{\boldsymbol{\theta}}_{t_j} - \hat{\boldsymbol{\theta}}_s\right\|^2 ds$$
$$\leq \frac{3Cc_d^2}{4} \int_{t_j}^{t_{j+1}} \left[\mathbb{E} \left\|-\nabla V(\hat{\boldsymbol{\theta}}_{\underline{s}}) + \alpha\phi[\pi_{M,c} * \hat{\rho}_{\underline{s}}](\hat{\boldsymbol{\theta}}_{\underline{s}})\right\|^2 (s - t_j)^2 + 2d(s - t_j)\right] ds$$
$$= Cc_d^2\eta^3 \sum_{j=0}^{k-1} \mathbb{E} \left\|-\nabla V(\hat{\boldsymbol{\theta}}_{t_j}) + \alpha\phi[\pi_{M,c} * \hat{\rho}_{t_j}](\hat{\boldsymbol{\theta}}_{t_j})\right\|^2 + Cc_d^2 dkh^2.$$

By the assumption that $\mathbb{E} \left\|\tilde{\boldsymbol{\theta}}_{t_j}\right\|$ is finite and $\tilde{\boldsymbol{\theta}}_{t_j} \stackrel{d}{=} \hat{\boldsymbol{\theta}}_{t_j}$, $\mathbb{E} \left\|\hat{\boldsymbol{\theta}}_{t_j}\right\|^2$ is also finite, we have

$$\mathbb{E} \left\|-\nabla V(\hat{\boldsymbol{\theta}}_{\underline{t}}) + \alpha\phi[\pi_{M,c} * \hat{\rho}_{\underline{t}}](\hat{\boldsymbol{\theta}}_{\underline{t}})\right\|^2$$
$$\leq 2\mathbb{E} \left\|\nabla V(\hat{\boldsymbol{\theta}}_{\underline{t}})\right\|^2 + 2\alpha^2\mathbb{E} \left\|\phi[\pi_{M,c} * \hat{\rho}_{\underline{t}}](\hat{\boldsymbol{\theta}}_{\underline{t}})\right\|^2$$
$$\leq 4b_1^2 + 4b_1^2\mathbb{E} \left\|\hat{\boldsymbol{\theta}}_{\underline{t}}\right\|^2 + 2\alpha^2\mathbb{E} \left(\left(\frac{2}{\sigma} + b_1\right) \mathbb{E}_{\boldsymbol{\theta}' \sim \pi_{M,c}*\hat{\rho}_{\underline{t}}} \|\boldsymbol{\theta}'\| + \|\boldsymbol{\theta}\|\right)^2$$
$$\leq c_d^2 C.$$

Thus we conclude that

$$I_1 + I_3 \leq Cc_d^2 \left(c_d^2 k\eta^3 + dk\eta^2\right) .$$

**Bounding $I_2$**

$$\mathbb{E}\left\|\phi[\pi_{M,c}*\hat{\rho}_{t_j}](\hat{\boldsymbol{\theta}}_{t_j}) - \phi[\pi_{M,c}*\bar{\rho}_s](\hat{\boldsymbol{\theta}}_{t_j})\right\|^2$$

$$=\mathbb{E}\left\|\frac{1}{M}\sum_{l=1}^{M}\left[\phi[\hat{\rho}_{t_j-cl}](\hat{\boldsymbol{\theta}}_{t_j}) - \phi[\bar{\rho}_{s-cl}](\hat{\boldsymbol{\theta}}_{t_j})\right]\right\|^2$$

$$\leq\frac{1}{M}\sum_{l=1}^{M}\mathbb{E}\left\|\phi[\hat{\rho}_{t_j-cl}](\hat{\boldsymbol{\theta}}_{t_j}) - \phi[\bar{\rho}_{s-cl}](\hat{\boldsymbol{\theta}}_{t_j})\right\|^2$$

$$=\frac{1}{M}\sum_{l=1}^{M}\mathbb{E}\left\|\mathbb{E}_{\boldsymbol{\theta}\sim\hat{\rho}_{t_j-cl}}\bar{\phi}_{\hat{\boldsymbol{\theta}}_{t_j}}(\boldsymbol{\theta}) - \mathbb{E}_{\boldsymbol{\theta}\sim\bar{\rho}_{s-cl}}\bar{\phi}_{\hat{\boldsymbol{\theta}}_{t_j}}(\boldsymbol{\theta})\right\|^2$$

$$=\frac{1}{M}\sum_{l=1}^{M}\mathbb{E}_{\hat{\boldsymbol{\theta}}_{t_j}}\sum_{i=1}^{d}\left|\mathbb{E}_{\boldsymbol{\theta}\sim\hat{\rho}_{t_j-cl}}\bar{\phi}_{\hat{\boldsymbol{\theta}}_{t_j},i}(\boldsymbol{\theta}) - \mathbb{E}_{\boldsymbol{\theta}\sim\bar{\rho}_{s-cl}}\bar{\phi}_{\hat{\boldsymbol{\theta}}_{t_j},i}(\boldsymbol{\theta})\right|^2$$

$$\leq\frac{1}{M}\sum_{l=1}^{M}\mathbb{E}_{\hat{\boldsymbol{\theta}}_{t_j}}\sum_{i=1}^{d}\left(\left\|\bar{\phi}_{\hat{\boldsymbol{\theta}}_{t_j},i}(\cdot)\right\|_{\mathcal{L}_\infty}\vee\left\|\bar{\phi}_{\hat{\boldsymbol{\theta}}_{t_j},i}(\cdot)\right\|_{\text{Lip}}\right)^2\mathbb{D}_{\text{BL}}^2\left[\hat{\rho}_{t_j-cl},\bar{\rho}_{s-cl}\right]$$

By Lemma E.4 and the Assumption 4.4 that $V$ is at most quadratic growth and that $\hat{\rho}_{\underline{t}}$ has finite second moment, we have

$$\mathbb{E}_{\hat{\boldsymbol{\theta}}_{t_j}}\sum_{i=1}^{d}\left(\left\|\bar{\phi}_{\hat{\boldsymbol{\theta}}_{t_j},i}(\cdot)\right\|_{\mathcal{L}_\infty}\vee\left\|\bar{\phi}_{\hat{\boldsymbol{\theta}}_{t_j},i}(\cdot)\right\|_{\text{Lip}}\right)^2$$

$$=\mathbb{E}_{\hat{\boldsymbol{\theta}}_{t_j}}\sum_{i=1}^{d}\left(\left\|\bar{\phi}_{\hat{\boldsymbol{\theta}}_{t_j},i}(\cdot)\right\|_{\mathcal{L}_\infty}^2\vee\left\|\bar{\phi}_{\hat{\boldsymbol{\theta}}_{t_j},i}(\cdot)\right\|_{\text{Lip}}^2\right)$$

$$\leq\left[4d\left\|\frac{2}{\sigma}e^{-\|\boldsymbol{\theta}\|^2/\sigma}\theta_1\right\|_{\text{BL}}^2 + 4\left\|e^{-\|\cdot\|^2/\sigma}\right\|_{\text{BL}}^2\mathbb{E}_{\hat{\boldsymbol{\theta}}_{t_j}}\left\|\nabla V(\hat{\boldsymbol{\theta}}_{t_j})\right\|^2\right]$$

$$\leq C(d+c_d^2).$$

Plug in the above estimation, we have

$$I_2 = \frac{3\alpha^2}{4}\sum_{j=0}^{k-1}\int_{t_j}^{t_{j+1}}\mathbb{E}\left\|\phi[\pi_{M,c}*\hat{\rho}_{t_j}](\hat{\boldsymbol{\theta}}_{t_j}) - \phi[\pi_{M,c}*\bar{\rho}_s](\hat{\boldsymbol{\theta}}_{t_j})\right\|^2 ds$$

$$\leq \alpha^2 C(d+c_d^2)\sum_{j=0}^{k-1}\int_{t_j}^{t_{j+1}}\frac{1}{M}\sum_{l=1}^{M}\mathbb{D}_{\text{BL}}^2\left[\hat{\rho}_{t_j-cl},\bar{\rho}_{s-cl}\right] ds$$

$$\leq \alpha^2 C(d+c_d^2)\sum_{j=0}^{k-1}\frac{1}{M}\sum_{l=1}^{M}\int_{t_j}^{t_{j+1}}\mathbb{D}_{\text{KL}}\left[\hat{\rho}_{t_j-cl},\bar{\rho}_{s-cl}\right] ds,$$

where the last inequality is due to the relation that $\mathbb{D}_{\text{BL}}^2\overset{\text{definition}}{\leq}\mathbb{D}_{\text{TV}}^2\overset{\text{Pinsker's}}{\leq}\mathbb{D}_{\text{KL}}$.

**Overall Bound** Combine all the estimation, we have

$$\mathbb{D}_{\text{KL}}\left[\bar{\rho}_{t_k}^{\boldsymbol{\theta}_0}\|\hat{\rho}_{t_k}^{\boldsymbol{\theta}_0}\right] \leq \alpha^2 C(d+c_d^2)\sum_{j=0}^{k-1}\frac{1}{M}\sum_{l=1}^{M}\int_{t_j}^{t_{j+1}}\mathbb{D}_{\text{KL}}\left[\hat{\rho}_{t_j-cl},\bar{\rho}_{s-cl}\right] ds + Cc_d^2\left(c_d^2k\eta^3 + dk\eta^2\right)$$

$$= \alpha^2 C(d+c_d^2)\sum_{j=0}^{k-1}\frac{1}{M}\sum_{l=1}^{M}\int_0^\eta\mathbb{D}_{\text{KL}}\left[\hat{\rho}_{t_{\left(\frac{j\eta-cl}{\eta}\right)}},\bar{\rho}_{t_{\left(\frac{j\eta-cl}{\eta}\right)+s}}\right] ds + Cc_d^2\left(c_d^2k\eta^3 + dk\eta^2\right)$$

Similar, if $k \le \frac{Mc}{\eta} - 1$, we have

$$\mathbb{D}_{\mathrm{KL}}\left[\bar{\rho}_{t_k}^{\boldsymbol{\theta}_0} \| \hat{\rho}_{t_k}^{\boldsymbol{\theta}_0}\right]$$

$$=\frac{1}{4}\int_0^{t_k} \mathbb{E}\left\|\nabla V(\hat{\boldsymbol{\theta}}_{\underline{s}}) - \nabla V(\hat{\boldsymbol{\theta}}_s)\right\|^2 ds$$

$$\le \frac{b_1^2}{4}\sum_{j=0}^{k-1}\int_{t_j}^{t_{j+1}} \mathbb{E}\left\|\hat{\boldsymbol{\theta}}_{t_j} - \hat{\boldsymbol{\theta}}_s\right\|^2 ds$$

$$\le \frac{b_1^2\eta^3}{12}\sum_{j=0}^{k-1} \mathbb{E}\left\|\nabla V(\hat{\boldsymbol{\theta}}_{t_j})\right\|^2 + \frac{dkb_1^2\eta^2}{4}$$

$$\le \frac{b_1^2\eta^3 k c_d^2}{12} + \frac{dkb_1^2\eta^2}{4}.$$

Define

$$u_k = \sup_{s\in[t_k,t_{k+1}]} \mathbb{D}_{\mathrm{KL}}\left[\bar{\rho}_{\underline{s}}^{\boldsymbol{\theta}_0} \| \hat{\rho}_s^{\boldsymbol{\theta}_0}\right],$$

and $U_k = \max_{l\in\{0,\dots,k\}} u_l$. We conclude that for $k \ge \frac{Mc}{\eta}$, for any $k' \le k$,

$$u_{k'} \le \alpha^2 C(d+c_d^2)\sum_{j=0}^{k-1}\frac{1}{M}\sum_{l=1}^{M}\int_0^h \mathbb{D}_{\mathrm{KL}}\left[\hat{\rho}_{t_{\left(\frac{j\eta-cl}{\eta}\right)}}, \bar{\rho}_{t_{\left(\frac{j\eta-cl}{\eta}\right)+s}}\right] ds + Cc_d^2\left(c_d^2 k\eta^3 + dk\eta^2\right)$$

$$\le \alpha^2 C(d+c_d^2)\sum_{j=0}^{k-1}\frac{1}{M}\sum_{l=1}^{M}\eta u_{\left(\frac{j\eta-cl}{\eta}\right)} + Cc_d^2\left(c_d^2 k\eta^3 + dk\eta^2\right)$$

$$\le \alpha^2 C(d+c_d^2)\eta\sum_{j=0}^{k-1}U_j + Cc_d^2\left(c_d^2 k\eta^3 + dk\eta^2\right).$$

For $k < \frac{Mc}{\eta}$, which is a simpler case, we have

$$U_k \le C\left(\eta^3 k c_d^2 + dk\eta^2\right) < CMc\left(\eta c_d^2 + d\right)\eta.$$

We bound the case when $k \ge \frac{Mc}{\eta}$,

$$U_k \le \alpha^2 C(d+c_d^2)\eta\sum_{j=0}^{k-1}U_j + Cc_d^2\left(c_d^2 k\eta^3 + dk\eta^2\right).$$

If we take $\eta$ sufficiently small, such that $c_d^2 k\eta^3 \le dk\eta^2$, we have

$$U_k \le \alpha^2 C(d+c_d^2)\eta\sum_{j=0}^{k-1}U_j + 2Cc_d^2 dk\eta^2$$

$$\le \alpha^2 C(d+c_d^2)\eta\sum_{j=0}^{k-1}(U_j + \eta).$$

Define $\eta' = \alpha^2 C(d+c_d^2)\eta$ and we can choose $\eta$ small enough such that $\eta' < 1/2$ and $\eta < 1/2$. Without loss of generality, we also assume $\eta' \ge \eta$ and thus we have

$$U_k \le \eta'\sum_{j=0}^{k-1}(U_j + \eta').$$

Also we assume $U_k \ge \eta'$, otherwise we conclude that $U_k < \eta'$. We thus have $U_k \le q\sum_{j=0}^{k-1}U_j$, where $q = 2\eta'$. Suppose that $U_{\frac{Mc}{\eta}-1} = x \le CMc\left(\eta c_d^2 + d\right)\eta$ and some algebra (which reduces to Pascal's triangle) shows that

$$U_k \le xq(1+q)^{k-\frac{Mc}{\eta}}.$$

We conclude that $U_k \leq xq(1+q)^{k-1}$. Notice that $q = 2\alpha^2 C(d+c_d^2)\eta$. Thus for any $k \geq Mc/\eta$,

$$
\begin{aligned}
U_k &\leq xq(1+q)^{k-\frac{Mc}{\eta}} \\
&= xq(1+q)^{(k\eta-Mc)/\eta} \\
&= xq(1+q)^{2\alpha^2 C(d+c_d^2)(k\eta-Mc)/q} \\
&\leq x2\alpha^2 C(d+c_d^2)e^{2\alpha^2 C(d+c_d^2)(k\eta-Mc)}\eta \\
&\leq CMc\alpha^2\left(\eta c_d^2 + d\right)(d+c_d^2)e^{2\alpha^2 C(d+c_d^2)(k\eta-Mc)}\eta^2,
\end{aligned}
$$

for sufficiently small $\eta$. Combine the above two estimations, we have

$$
U_k \leq \begin{cases} C\left(\eta^3 kc_d^2 + dk\eta^2 + \eta\right) & k \leq Mc/\eta - 1 \\ CMc\alpha^2\left(\eta c_d^2 + d\right)(d+c_d^2)e^{2\alpha^2 C(d+c_d^2)(k\eta-Mc)}\eta^2 + C\eta & k \geq Mc/\eta \end{cases}.
$$

Notice that now we have $U_k = \max_{l\in\{0,\ldots,k\}} \sup_{s\in[0,\eta]} \mathbb{D}_{\mathrm{KL}}\left[\bar{\rho}_{l\eta+s}^{\boldsymbol{\theta}_0} \| \tilde{\rho}_{l\eta}^{\boldsymbol{\theta}_0}\right]$, which is a function of $\boldsymbol{\theta}_0$. We then bound $\bar{U}_k = \max_{l\in\{0,\ldots,k\}} \sup_{s\in[0,\eta]} \mathbb{D}_{\mathrm{KL}}\left[\bar{\rho}_{l\eta+s} \| \tilde{\rho}_{l\eta}\right]$. Notice that the KL divergence has the following variational representation:

$$
\mathbb{D}_{\mathrm{KL}}[\rho_1 \| \rho_2] = \sup_f \left[\mathbb{E}_{\rho_1} f - \mathbb{E}_{\rho_2} e^f\right],
$$

where the $f$ is chosen in the set that $\mathbb{E}_{\rho_1} f$ and $\mathbb{E}_{\rho_2} e^f$ exist. And thus we have

$$
\begin{aligned}
\mathbb{D}_{\mathrm{KL}}[\bar{\rho}_{l\eta+s} \| \tilde{\rho}_{l\eta}] &= \sup_f \left[\mathbb{E}_{\boldsymbol{\theta}_0\sim\rho_0}\left(\mathbb{E}_{\bar{\rho}_{l\eta+s}^{\boldsymbol{\theta}_0}} f - \mathbb{E}_{\tilde{\rho}_{l\eta}^{\boldsymbol{\theta}_0}} e^f\right)\right] \\
&\leq \mathbb{E}_{\boldsymbol{\theta}_0\sim\rho_0} \sup_f \left[\left(\mathbb{E}_{\bar{\rho}_{l\eta+s}^{\boldsymbol{\theta}_0}} f - \mathbb{E}_{\tilde{\rho}_{l\eta}^{\boldsymbol{\theta}_0}} e^f\right)\right].
\end{aligned}
$$

And thus $\bar{U}_k \leq U_k$. Also the inequality that

$$
\bar{U}_k = \max_{l\in\{0,\ldots,k\}} \sup_{s\in[0,\eta]} \mathbb{D}_{\mathrm{KL}}\left[\bar{\rho}_{l\eta+s} \| \tilde{\rho}_{l\eta}\right] \geq \max_{l\in\{0,\ldots,k\}} \mathbb{D}_{\mathrm{KL}}\left[\bar{\rho}_{l\eta} \| \tilde{\rho}_{l\eta}\right]
$$

holds naturally by definition. We complete the proof.

### E.5.5 Proof of Theorem 4.3

The constant $h_1$ is defined as

$$
h_1 = \left\|\frac{2}{\sigma}e^{-\|\boldsymbol{\theta}\|^2/\sigma}\boldsymbol{\theta}_1\right\|_{\mathrm{BL}}^2 \vee \left\|e^{-\|\cdot\|^2/\sigma}\right\|_{\mathrm{BL}}^2 \vee \left\|\frac{2}{\sigma}e^{-(\cdot)^2/\sigma}(\cdot)\right\|_{\mathrm{Lip}}.
$$

Now we start the proof. We couple the process of $\boldsymbol{\theta}_k$ and $\tilde{\boldsymbol{\theta}}_k$ by the same gaussian noise $e_k$ in every iteration and same initialization $\tilde{\boldsymbol{\theta}}_0 = \boldsymbol{\theta}_0$. For $k \leq Mc/\eta-1$, $\mathbb{E}\left\|\boldsymbol{\theta}_k - \tilde{\boldsymbol{\theta}}_k\right\|^2 = 0$ and for $k \geq Mc/\eta$ we have the following inequality,

$$
\begin{aligned}
&\mathbb{E}\left\|\boldsymbol{\theta}_{k+1} - \tilde{\boldsymbol{\theta}}_{k+1}\right\|^2 - \mathbb{E}\left\|\boldsymbol{\theta}_k - \tilde{\boldsymbol{\theta}}_k\right\|^2 \\
=&2\eta\mathbb{E}\left\langle\boldsymbol{\theta}_k - \tilde{\boldsymbol{\theta}}_k, -\nabla V(\boldsymbol{\theta}_k) + \nabla V(\tilde{\boldsymbol{\theta}}_k)\right\rangle \\
&+2\eta\alpha\mathbb{E}\left\langle\boldsymbol{\theta}_k - \tilde{\boldsymbol{\theta}}_k, \phi[\frac{1}{M}\sum_{j=1}^M \delta_{\boldsymbol{\theta}_{k-jc/\eta}}](\boldsymbol{\theta}_k) - \phi[\pi_{M,c/\eta} * \tilde{\rho}_k](\tilde{\boldsymbol{\theta}}_k)\right\rangle \\
&+\eta^2\mathbb{E}\left\|-\nabla V(\boldsymbol{\theta}_k) + \alpha\phi[\frac{1}{M}\sum_{j=1}^M \delta_{\boldsymbol{\theta}_{k-jc/\eta}}](\boldsymbol{\theta}_k) + \nabla V(\tilde{\boldsymbol{\theta}}_k) - \alpha\phi[\pi_{M,c/\eta} * \tilde{\rho}_k](\tilde{\boldsymbol{\theta}}_k)\right\|^2.
\end{aligned}
$$

By the log-concavity, we have

$$\mathbb{E}\left\langle \boldsymbol{\theta}_k - \tilde{\boldsymbol{\theta}}_k, -\nabla V(\boldsymbol{\theta}_k) + \nabla V(\tilde{\boldsymbol{\theta}}_k) \right\rangle$$
$$\leq -L\mathbb{E}\left\| \boldsymbol{\theta}_k - \tilde{\boldsymbol{\theta}}_k \right\|^2,$$

for some positive constant $L$. And also, as $\eta$ is small, the last term on the right side of the equation is small term. Thus our main target is to bound the second term. We decompose the second term on the left side of the equation by

$$\mathbb{E}\left\langle \boldsymbol{\theta}_k - \tilde{\boldsymbol{\theta}}_k, \phi[\frac{1}{M}\sum_{j=1}^{M}\delta_{\boldsymbol{\theta}_{k-jc}}](\boldsymbol{\theta}_k) - \phi[\pi_{M,c/\eta} * \tilde{\rho}_k](\tilde{\boldsymbol{\theta}}_k) \right\rangle$$
$$= \mathbb{E}\left\langle \boldsymbol{\theta}_k - \tilde{\boldsymbol{\theta}}_k, \phi[\frac{1}{M}\sum_{j=1}^{M}\delta_{\boldsymbol{\theta}_{k-jc/\eta}}](\boldsymbol{\theta}_k) - \phi[\pi_{M,c/\eta} * \rho_k](\boldsymbol{\theta}_k) \right\rangle$$
$$+ \mathbb{E}\left\langle \boldsymbol{\theta}_k - \tilde{\boldsymbol{\theta}}_k, \phi[\pi_{M,c/\eta} * \rho_k](\boldsymbol{\theta}_k) - \phi[\pi_{M,c/\eta} * \tilde{\rho}_k](\boldsymbol{\theta}_k) \right\rangle$$
$$+ \mathbb{E}\left\langle \boldsymbol{\theta}_k - \tilde{\boldsymbol{\theta}}_k, \phi[\pi_{M,c/\eta} * \tilde{\rho}_k](\boldsymbol{\theta}_k) - \phi[\pi_{M,c/\eta} * \tilde{\rho}_k](\tilde{\boldsymbol{\theta}}_k) \right\rangle$$
$$= I_1 + I_2 + I_3.$$

We bound $I_1$, $I_2$ and $I_3$ independently.

**Bounding $I_1$**

By Holder's inequality,

$$I_1 \leq \mathbb{E}\left[ \left\| \boldsymbol{\theta}_k - \tilde{\boldsymbol{\theta}}_k \right\| \left\| \phi[\frac{1}{M}\sum_{j=1}^{M}\delta_{\boldsymbol{\theta}_{k-jc/\eta}}](\boldsymbol{\theta}_k) - \phi[\pi_{M,c/\eta} * \rho_k](\boldsymbol{\theta}_k) \right\| \right]$$
$$\leq \sqrt{\mathbb{E}\left\| \boldsymbol{\theta}_k - \tilde{\boldsymbol{\theta}}_k \right\|^2} \sqrt{\mathbb{E}\left\| \phi[\frac{1}{M}\sum_{j=1}^{M}\delta_{\boldsymbol{\theta}_{k-jc/\eta}}](\boldsymbol{\theta}_k) - \phi[\pi_{M,c/\eta} * \rho_k](\boldsymbol{\theta}_k) \right\|^2}.$$

We bound the second term on the right side of the inequality. Define

$$a_2 = \sup_k \frac{\mathbb{E}\left\| \phi[\frac{1}{M}\sum_{j=1}^{M}\delta_{\boldsymbol{\theta}_{k-jc/\eta}}](\boldsymbol{\theta}_k) - \phi[\pi_{M,c/\eta} * \rho_k](\boldsymbol{\theta}_k) \right\|^2}{\sup_{\|\boldsymbol{\theta}\|\leq B}\mathbb{E}\left\| \phi[\frac{1}{M}\sum_{j=1}^{M}\delta_{\boldsymbol{\theta}_{k-jc/\eta}}](\boldsymbol{\theta}) - \phi[\pi_{M,c/\eta} * \rho_k](\boldsymbol{\theta}) \right\|^2}$$

and by the regularity assumption we know that $a_2 < \infty$. Define $\phi[\frac{1}{M}\sum_{j=1}^{M}\delta_{\boldsymbol{\theta}_{k-jc/\eta}}](\boldsymbol{\theta}) - \phi[\pi_{M,c/\eta} * \rho_k](\boldsymbol{\theta}) = \phi^*[\frac{1}{M}\sum_{j=1}^{M}\delta_{\boldsymbol{\theta}_{k-jc/\eta}}]$ and since the stein operator is linear functional of the distribution, we have

$$\mathbb{E}\phi^*[\frac{1}{M}\sum_{j=1}^{M}\delta_{\boldsymbol{\theta}_{k-jc/\eta}}](\boldsymbol{\theta}) = 0,$$

given any $\boldsymbol{\theta}$. By Theorem E.1 that $\Theta_k$ is geometric ergodicity and thus is $\beta$-mixing with exponentially fast decay rate by Proposition E.2. And by Proposition E.1, we know that $\Theta_k$ is also $\alpha$-mixing

with exponentially fast decay rate. We have the following estimation

$$\mathbb{E}\left\|\phi[\frac{1}{M}\sum_{j=1}^{M}\delta_{\boldsymbol{\theta}_{k-jc/\eta}}](\boldsymbol{\theta}_k) - \phi[\pi_{M,c/\eta}*\rho_k](\boldsymbol{\theta}_k)\right\|^2$$

$$\leq a_2 \sup_{\|\boldsymbol{\theta}\|\leq B}\mathbb{E}\left\|\phi^*[\frac{1}{M}\sum_{j=1}^{M}\delta_{\boldsymbol{\theta}_{k-jc/\eta}}](\boldsymbol{\theta})\right\|^2$$

$$\leq \frac{a_2}{M^2}\sup_{\|\boldsymbol{\theta}\|\leq B}\mathbb{E}\sum_{k=1}^{M}\left\|\phi^*[\delta_{\boldsymbol{\theta}_{t-kc/\eta}}](\boldsymbol{\theta})\right\|^2$$

$$+\frac{a_2}{M^2}\sup_{\|\boldsymbol{\theta}\|\leq B}\mathbb{E}\sum_{k\neq j}\left\langle\phi^*[\delta_{\boldsymbol{\theta}_{t-kc/\eta}}](\boldsymbol{\theta}),\phi^*[\delta_{\boldsymbol{\theta}_{t-jc/\eta}}](\boldsymbol{\theta})\right\rangle$$

$$\leq \frac{Ca_2}{M}\left[\frac{e^{-rc}\left(1-e^{-rMc}\right)}{1-e^{rc}}+1\right],$$

for some positive constant $r$ that characterize the decay rate of $\alpha$ mixing. Notice that here $\eta$ is canceled because the decay rate of mixing is $\mathcal{O}(\eta)$ (on the power of exponential) and $c/\eta = \mathcal{O}(\eta^{-1})$. Combine this two estimations, we have

$$I_1 \leq \sqrt{\mathbb{E}\left\|\boldsymbol{\theta}_k - \tilde{\boldsymbol{\theta}}_k\right\|^2}\sqrt{\frac{a_2C}{M}\left[\frac{e^{-rc}\left(1-e^{-rMc}\right)}{1-e^{rc}}+1\right]}.$$

**Bounding $I_2$** By Holder's inequality, we have

$$I_2 \leq \sqrt{\mathbb{E}\left\|\boldsymbol{\theta}_k - \tilde{\boldsymbol{\theta}}_k\right\|^2}\sqrt{\mathbb{E}\left\|\phi[\pi_{M,c/\eta}*\rho_k](\boldsymbol{\theta}_k) - \phi[\pi_{M,c/\eta}*\tilde{\rho}_k](\boldsymbol{\theta}_k)\right\|^2}.$$

We bound the second term in the right side of the inequality.

$$\mathbb{E}\left\|\phi[\pi_{M,c/\eta}*\rho_k](\boldsymbol{\theta}_k) - \phi[\pi_{M,c/\eta}*\tilde{\rho}_k](\boldsymbol{\theta}_k)\right\|^2$$

$$=\mathbb{E}\left\|\frac{1}{M}\sum_{j=1}^{M}\left[\phi[\rho_{k-jc/\eta}](\boldsymbol{\theta}_k) - \phi[\tilde{\rho}_{k-jc/\eta}](\boldsymbol{\theta}_k)\right]\right\|^2$$

$$\leq \frac{1}{M}\sum_{j=1}^{M}\mathbb{E}\left\|\phi[\rho_{k-jc/\eta}](\boldsymbol{\theta}_k) - \phi[\tilde{\rho}_{k-jc/\eta}](\boldsymbol{\theta}_k)\right\|^2$$

$$=\frac{1}{M}\sum_{j=1}^{M}\mathbb{E}_{\boldsymbol{\theta}_k}\left\|\mathbb{E}_{\boldsymbol{\theta}\sim\rho_{k-jc/\eta}}\bar{\phi}_{\boldsymbol{\theta}_k}(\boldsymbol{\theta}) - \mathbb{E}_{\boldsymbol{\theta}\sim\tilde{\rho}_{k-jc/\eta}}\bar{\phi}_{\boldsymbol{\theta}_k}(\boldsymbol{\theta})\right\|^2$$

$$=\frac{1}{M}\sum_{j=1}^{M}\mathbb{E}_{\boldsymbol{\theta}_k}\sum_{i=1}^{d}\left|\mathbb{E}_{\boldsymbol{\theta}\sim\rho_{k-jc/\eta}}\bar{\phi}_{\boldsymbol{\theta}_k,i}(\boldsymbol{\theta}) - \mathbb{E}_{\boldsymbol{\theta}\sim\tilde{\rho}_{k-jc/\eta}}\bar{\phi}_{\boldsymbol{\theta}_k,i}(\boldsymbol{\theta})\right|^2$$

$$\leq \frac{1}{M}\sum_{j=1}^{M}\mathbb{E}_{\boldsymbol{\theta}_k}\sum_{i=1}^{d}\left(\left\|\bar{\phi}_{\boldsymbol{\theta}_k,i}(\cdot)\right\|_{\mathcal{L}_\infty}\vee\left\|\bar{\phi}_{\boldsymbol{\theta}_k,i}(\cdot)\right\|_{\text{Lip}}\right)^2\mathbb{D}_{\text{BL}}^2\left[\rho_{k-jc/\eta},\tilde{\rho}_{k-jc/\eta}\right].$$

By Lemma E.4, we have

$$\sum_{i=1}^{d}\left(\left\|\bar{\phi}_{\boldsymbol{\theta}_k,i}(\cdot)\right\|_{\mathcal{L}_\infty}\vee\left\|\bar{\phi}_{\boldsymbol{\theta}_k,i}(\cdot)\right\|_{\text{Lip}}\right)^2$$

$$=\sum_{i=1}^{d}\left(\left\|\bar{\phi}_{\boldsymbol{\theta}_k,i}(\cdot)\right\|_{\mathcal{L}_\infty}^2\vee\left\|\bar{\phi}_{\boldsymbol{\theta}_k,i}(\cdot)\right\|_{\text{Lip}}^2\right)$$

$$\leq \left[4d\left\|\frac{2}{\sigma}e^{-\|\boldsymbol{\theta}\|^2/\sigma}\theta_1\right\|_{\text{BL}}^2 + 4\left\|e^{-\|\cdot\|^2/\sigma}\right\|_{\text{BL}}^2\|\nabla V(\boldsymbol{\theta}_k)\|^2\right].$$

Plug in the above estimation and by the relation that $\mathbb{D}_{\mathrm{BL}} \le \mathbb{W}_1 \le \mathbb{W}_2$, we have

$$\mathbb{E}\left\| \phi[\pi_{M,c} * \rho_k](\boldsymbol{\theta}_k) - \phi[\pi_{M,c} * \tilde{\rho}_k](\boldsymbol{\theta}_k) \right\|^2$$

$$\le \left[ 4d \left\| \frac{2}{\sigma} e^{-\|\boldsymbol{\theta}\|^2/\sigma} \theta_1 \right\|_{\mathrm{BL}}^2 + 4 \left\| e^{-\|\cdot\|^2/\sigma} \right\|_{\mathrm{BL}}^2 \mathbb{E}_{\boldsymbol{\theta}_k} \|\nabla V(\boldsymbol{\theta}_k)\|^2 \right] \frac{1}{M} \sum_{j=1}^M \mathbb{D}_{\mathrm{BL}}^2 \left[ \rho_{k-cj}, \rho_{\tilde{k}-cj} \right]$$

$$\le \left[ 4d \left\| \frac{2}{\sigma} e^{-\|\boldsymbol{\theta}\|^2/\sigma} \theta_1 \right\|_{\mathrm{BL}}^2 + 4 \left\| e^{-\|\cdot\|^2/\sigma} \right\|_{\mathrm{BL}}^2 \mathbb{E}_{\boldsymbol{\theta}_k} \|\nabla V(\boldsymbol{\theta}_k)\|^2 \right] \frac{1}{M} \sum_{j=1}^M \mathbb{W}_2^2 \left[ \rho_{k-cj}, \rho_{\tilde{k}-cj} \right].$$

And combined all the estimation and by the definition of Wasserstein-distance, we conclude that

$$I_2 \le \sqrt{ 4d \left\| \frac{2}{\sigma} e^{-\|\boldsymbol{\theta}\|^2/\sigma} \theta_1 \right\|_{\mathrm{BL}}^2 + 4 \left\| e^{-\|\cdot\|^2/\sigma} \right\|_{\mathrm{BL}}^2 \mathbb{E}_{\boldsymbol{\theta}_k} \|\nabla V(\boldsymbol{\theta}_k)\|^2 } \sqrt{ \frac{1}{M} \sum_{j=1}^M \mathbb{W}_2^2 \left[ \rho_{k-cj}, \tilde{\rho}_{k-cj} \right] }$$

$$\le \sqrt{ 4d \left\| \frac{2}{\sigma} e^{-\|\boldsymbol{\theta}\|^2/\sigma} \theta_1 \right\|_{\mathrm{BL}}^2 + 4 \left\| e^{-\|\cdot\|^2/\sigma} \right\|_{\mathrm{BL}}^2 \mathbb{E}_{\boldsymbol{\theta}_k} \|\nabla V(\boldsymbol{\theta}_k)\|^2 } \sqrt{ \frac{1}{M} \sum_{j=1}^M \mathbb{E} \left\| \boldsymbol{\theta}_{k-cj} - \tilde{\boldsymbol{\theta}}_{k-cj} \right\|^2 }.$$

**Bounding $I_3$**

By Holder's inequality,

$$I_3 \le \sqrt{ \mathbb{E} \left\| \boldsymbol{\theta}_k - \tilde{\boldsymbol{\theta}}_k \right\|^2 } \sqrt{ \mathbb{E} \left\| \phi[\pi_{M,c/\eta} * \tilde{\rho}_k](\boldsymbol{\theta}_k) - \phi[\pi_{M,c/\eta} * \tilde{\rho}_k](\tilde{\boldsymbol{\theta}}_k) \right\|^2 }.$$

We bound the last term on the right side of the inequality. By assumption and Lemma E.3, we have

$$\mathbb{E} \left\| \phi[\pi_{M,c/\eta} * \tilde{\rho}_k](\boldsymbol{\theta}_k) - \phi[\pi_{M,c/\eta} * \tilde{\rho}_k](\tilde{\boldsymbol{\theta}}_k) \right\|^2$$

$$\le \left[ \left\| e^{-(\cdot)^2/\sigma} \right\|_{\mathrm{Lip}} \mathbb{E}_{\boldsymbol{\theta} \sim \tilde{\rho}_k} \|\nabla V(\boldsymbol{\theta})\| + \left\| \frac{2}{\sigma} e^{-(\cdot)^2/\sigma}(\cdot) \right\|_{\mathrm{Lip}} \right]^2 \mathbb{E} \left\| \boldsymbol{\theta}_k - \tilde{\boldsymbol{\theta}}_k \right\|^2.$$

And combine the estimation, we have

$$I_3 \le \left[ \left\| e^{-(\cdot)^2/\sigma} \right\|_{\mathrm{Lip}} \mathbb{E}_{\boldsymbol{\theta} \sim \tilde{\rho}_k} \|\nabla V(\boldsymbol{\theta})\| + \left\| \frac{2}{\sigma} e^{-(\cdot)^2/\sigma}(\cdot) \right\|_{\mathrm{Lip}} \right] \mathbb{E} \left\| \boldsymbol{\theta}_k - \tilde{\boldsymbol{\theta}}_k \right\|^2.$$

**Overall Bound** Combine all the results, we have the following bound: for $k \ge Mc$,

$$\mathbb{E} \left\| \boldsymbol{\theta}_{k+1} - \tilde{\boldsymbol{\theta}}_{k+1} \right\|^2 - \mathbb{E} \left\| \boldsymbol{\theta}_k - \tilde{\boldsymbol{\theta}}_k \right\|^2$$

$$\le - 2\eta L \mathbb{E} \left\| \boldsymbol{\theta}_k - \tilde{\boldsymbol{\theta}}_k \right\|^2$$

$$+ 2\eta\alpha \sqrt{ \mathbb{E} \left\| \boldsymbol{\theta}_k - \tilde{\boldsymbol{\theta}}_k \right\|^2 } \frac{c_1}{\sqrt{M}}$$

$$+ 2\eta\alpha c_2 \sqrt{ \frac{1}{M} \sum_{j=1}^M \mathbb{E} \left\| \boldsymbol{\theta}_{k-jc/\eta} - \tilde{\boldsymbol{\theta}}_{k-jc/\eta} \right\|^2 \mathbb{E} \left\| \boldsymbol{\theta}_k - \tilde{\boldsymbol{\theta}}_k \right\|^2 }$$

$$+ 2\eta\alpha c_3 \mathbb{E} \left\| \boldsymbol{\theta}_k - \tilde{\boldsymbol{\theta}}_k \right\|^2$$

$$+ \eta^2 c_4,$$

where

$$c_1 = \sqrt{ a_2 C \left[ \frac{e^{-rc}\left(1 - e^{-rMc}\right)}{1 - e^{rc}} + 1 \right] },$$

$$c_2 = \sqrt{4d \left\| \frac{2}{\sigma} e^{-\|\boldsymbol{\theta}\|^2/\sigma} \theta_1 \right\|_{\mathrm{BL}}^2 + 4 \left\| e^{-\|\cdot\|^2/\sigma} \right\|_{\mathrm{BL}}^2 \sup_k \mathbb{E}_{\boldsymbol{\theta}_k} \|\nabla V(\boldsymbol{\theta}_k)\|^2},$$

$$c_3 = \left[ \left\| e^{-(\cdot)^2/\sigma} \right\|_{\mathrm{Lip}} \sup_k \mathbb{E}_{\boldsymbol{\theta} \sim \tilde{\rho}_k} \|\nabla V(\boldsymbol{\theta})\| + \left\| \frac{2}{\sigma} e^{-(\cdot)^2/\sigma}(\cdot) \right\|_{\mathrm{Lip}} \right],$$

and

$$c_4 = \sup_{k \geq Mc/\eta} \mathbb{E} \left\| \nabla V(\boldsymbol{\theta}_k) + \alpha \phi\left[ \frac{1}{M} \sum_{j=1}^{M} \delta_{\boldsymbol{\theta}_{k-jc/\eta}} \right](\boldsymbol{\theta}_k) - \nabla V(\tilde{\boldsymbol{\theta}}_k) - \alpha \phi[\pi_{M,c/\eta} * \tilde{\rho}_k](\tilde{\boldsymbol{\theta}}_k) \right\|^2.$$

Define $u_k = \sqrt{\mathbb{E}\left\| \boldsymbol{\theta}_k - \tilde{\boldsymbol{\theta}}_k \right\|^2}$ and $U_k = \sup_{l \in [k]} u_l$, we have

$$U_{k+1}^2 \leq q U_k^2 + \frac{2\eta\alpha c_1}{\sqrt{M}} U_k + \eta^2 c_4,$$

where $q = (1 - 2\eta(L - \alpha c_2 - \alpha c_3))$. By the assumption that $\alpha \leq L/(c_2 + c_3)$, $q < 1$. Now we prove the bound of $U_k$ by induction. We take the hypothesis that $U_k^2 \leq \frac{\left( \frac{2\eta\alpha c_1}{\sqrt{M}} + (1-q)\eta\left(c_4 + \frac{1}{1-q}\right) \right)^2}{(1-q)^2}$ and notice that the hypothesis holds for $U_0 = 0$. By the hypothesis, we have

$$U_{k+1}^2 \leq q \frac{\left( \frac{2\eta\alpha c_1}{\sqrt{M}} + (1-q)\eta\left(c_4 + \frac{1}{1-q}\right) \right)^2}{(1-q)^2} + \frac{2\eta\alpha c_1}{\sqrt{M}} \frac{\left( \frac{2\eta\alpha c_1}{\sqrt{M}} + (1-q)\eta\left(c_4 + \frac{1}{1-q}\right) \right)}{(1-q)} + \eta^2\left(c_4 + \frac{1}{1-q}\right)$$

$$= q \frac{\left( \frac{2\eta\alpha c_1}{\sqrt{M}} + (1-q)\eta\left(c_4 + \frac{1}{1-q}\right) \right)^2}{(1-q)^2} + \frac{1}{1-q}\left( \frac{2\eta\alpha c_1}{\sqrt{M}} \right)^2 + \frac{2\eta\alpha c_1}{\sqrt{M}}\eta\left(c_4 + \frac{1}{1-q}\right) + \eta^2\left(c_4 + \frac{1}{1-q}\right)$$

$$= q \frac{\left( \frac{2\eta\alpha c_1}{\sqrt{M}} + (1-q)\eta\left(c_4 + \frac{1}{1-q}\right) \right)^2}{(1-q)^2}$$

$$+ \frac{1-q}{(1-q)^2}\left[ \left( \frac{2\eta\alpha c_1}{\sqrt{M}} \right)^2 + (1-q)\frac{2\eta\alpha c_1}{\sqrt{M}}\eta\left(c_4 + \frac{1}{1-q}\right) + (1-q)\eta^2\left(c_4 + \frac{1}{1-q}\right) \right]$$

$$\leq q \frac{\left( \frac{2\eta\alpha c_1}{\sqrt{M}} + (1-q)\eta\left(c_4 + \frac{1}{1-q}\right) \right)^2}{(1-q)^2}$$

$$+ \frac{1-q}{(1-q)^2}\left[ \left( \frac{2\eta\alpha c_1}{\sqrt{M}} \right)^2 + (1-q)\frac{2\eta\alpha c_1}{\sqrt{M}}\eta\left(c_4 + \frac{1}{1-q}\right) + (1-q)^2\eta^2\left(c_4 + \frac{1}{1-q}\right)^2 \right]$$

$$= q \frac{\left( \frac{2\eta\alpha c_1}{\sqrt{M}} + (1-q)\eta\left(c_4 + \frac{1}{1-q}\right) \right)^2}{(1-q)^2} + (1-q)\frac{\left( \frac{2\eta\alpha c_1}{\sqrt{M}} + (1-q)\eta\left(c_4 + \frac{1}{1-q}\right) \right)^2}{(1-q)^2}$$

$$= \frac{\left( \frac{2\eta\alpha c_1}{\sqrt{M}} + (1-q)\eta\left(c_4 + \frac{1}{1-q}\right) \right)^2}{(1-q)^2},$$

where the last second inequality holds by $(1 - q)\left(c_4 + \frac{1}{1-q}\right) \geq 1$. Thus we complete the argument of induction and we have, for any $k$,

$$
U_k^2 \leq \frac{\left(\frac{2\eta\alpha c_1}{\sqrt{M}} + (1-q)\eta\left(c_4 + \frac{1}{1-q}\right)\right)^2}{(1-q)^2}
$$

$$
\leq 2\frac{\frac{4\eta^2\alpha^2 c_1^2}{M} + 2(1-q)^2\eta^2\left(c_4 + \frac{1}{1-q}\right)^2}{(1-q)^2}
$$

$$
= \frac{2\alpha^2 c_1^2}{(L - \alpha c_2 - \alpha c_3)^2}\frac{1}{M} + 4\eta^2\left(c_4 + 2\eta(L - \alpha c_2 - \alpha c_3)\right)^2.
$$

And it implies that $\mathbb{W}_2^2[\rho_k, \tilde{\rho}_k] \leq u_k \leq U_k \leq \frac{2\alpha^2 c_1^2}{(L-\alpha c_2 - \alpha c_3)^2}\frac{1}{M} + 4\eta^2\left(c_4 + 2\eta(L - \alpha c_2 - \alpha c_3)\right)^2.$

### E.6 Proof of Technical Lemmas

#### E.6.1 Proof of Lemma E.1

For the first part:

$$
\|\nabla V(\boldsymbol{\theta})\|
$$
$$
\leq \|\nabla V(\boldsymbol{\theta}) - \nabla V(\mathbf{0})\| + \|\nabla V(\mathbf{0})\|
$$
$$
\leq b_1\left(\|\boldsymbol{\theta}_1\| + 1\right).
$$

For the second part:

$$
\|\boldsymbol{\theta} - \eta\nabla V(\boldsymbol{\theta})\|
$$
$$
= \langle \boldsymbol{\theta} - \eta\nabla V(\boldsymbol{\theta}), \boldsymbol{\theta} - \eta\nabla V(\boldsymbol{\theta})\rangle
$$
$$
= \|\boldsymbol{\theta}\|^2 + 2\eta\langle \boldsymbol{\theta}, -\nabla V(\boldsymbol{\theta})\rangle + \eta^2\|\nabla V(\boldsymbol{\theta})\|^2
$$
$$
\leq \|\boldsymbol{\theta}\|^2 + 2\eta\left(-a_1\|\boldsymbol{\theta}\|^2 + b_1\right) + \eta^2 b_1(1 + \|\boldsymbol{\theta}\|^2)
$$
$$
= \left(1 - 2\eta a_1 + \eta^2 b_1\right)\|\boldsymbol{\theta}\|^2 + \eta^2 b_1 + 2\eta b_1.
$$

#### E.6.2 Proof of Lemma E.2

It is obvious that $\|K\|_{\infty,\infty} \leq 1$.

$$
\|K(\boldsymbol{\theta}', \boldsymbol{\theta}_1) - K(\boldsymbol{\theta}', \boldsymbol{\theta}_2)\|
$$
$$
\left\|e^{-\|\boldsymbol{\theta}'-\boldsymbol{\theta}_1\|^2/\sigma} - e^{-\|\boldsymbol{\theta}'-\boldsymbol{\theta}_2\|^2/\sigma}\right\|
$$
$$
\leq \left\|e^{-(\cdot)^2/\sigma}\right\|_{\text{Lip}}\|\boldsymbol{\theta}_1 - \boldsymbol{\theta}_2\|_2.
$$

And

$$
\|\nabla_{\boldsymbol{\theta}'}K(\boldsymbol{\theta}', \boldsymbol{\theta}_1) - \nabla_{\boldsymbol{\theta}'}K(\boldsymbol{\theta}', \boldsymbol{\theta}_2)\|
$$
$$
= \left\|\frac{2}{\sigma}e^{-\|\boldsymbol{\theta}'-\boldsymbol{\theta}_1\|^2/\sigma}(\boldsymbol{\theta}' - \boldsymbol{\theta}_1) - \frac{2}{\sigma}e^{-\|\boldsymbol{\theta}'-\boldsymbol{\theta}_2\|^2/\sigma}(\boldsymbol{\theta}' - \boldsymbol{\theta}_2)\right\|
$$
$$
\leq \left\|\frac{2}{\sigma}e^{-(\cdot)^2/\sigma}(\cdot)\right\|_{\text{Lip}}\|\boldsymbol{\theta}_1 - \boldsymbol{\theta}_2\|_2.
$$

### E.6.3 Proof of Lemma E.3

For any distribution $\rho$ such that $\int \|\nabla_{\boldsymbol{\theta}} V(\boldsymbol{\theta})\| \rho(\boldsymbol{\theta}) d\boldsymbol{\theta} < \infty$,

$$\|\phi[\rho](\boldsymbol{\theta}_1) - \phi[\rho](\boldsymbol{\theta}_2)\|$$

$$= \|\mathbb{E}_{\boldsymbol{\theta} \sim \rho} \{ - [K(\boldsymbol{\theta}, \boldsymbol{\theta}_1) - K(\boldsymbol{\theta}, \boldsymbol{\theta}_2)] \nabla V(\boldsymbol{\theta}) + \nabla_1 K(\boldsymbol{\theta}, \boldsymbol{\theta}_1) - \nabla_1 K(\boldsymbol{\theta}, \boldsymbol{\theta}_2) \} \|$$

$$\leq \left\| e^{-(\cdot)^2/\sigma} \right\|_{\text{Lip}} \mathbb{E}_{\boldsymbol{\theta} \sim \rho} \|\nabla V(\boldsymbol{\theta})\| \|\boldsymbol{\theta}_1 - \boldsymbol{\theta}_2\|_2$$

$$+ \left\| \frac{2}{\sigma} e^{-(\cdot)^2/\sigma}(\cdot) \right\|_{\text{Lip}} \|\boldsymbol{\theta}_1 - \boldsymbol{\theta}_2\|_2.$$

For proving the second result, we notice that

$$\|\phi[\rho](\boldsymbol{\theta})\| = \mathbb{E}_{\boldsymbol{\theta}' \sim \rho} [K(\boldsymbol{\theta}', \boldsymbol{\theta}) \nabla V(\boldsymbol{\theta}') + \nabla_1 K(\boldsymbol{\theta}', \boldsymbol{\theta})]$$

$$\leq \|K\|_\infty \mathbb{E}_{\boldsymbol{\theta}' \sim \rho} \left[ \|\nabla V(\boldsymbol{\theta}')\| + \frac{2}{\sigma} (\|\boldsymbol{\theta}'\| + \|\boldsymbol{\theta}\|) \right]$$

$$\leq \|K\|_\infty b_1 + \mathbb{E}_{\boldsymbol{\theta}' \sim \rho} \left[ \left( \frac{2}{\sigma} + b_1 \right) \|\boldsymbol{\theta}'\| + \|\boldsymbol{\theta}\| \right].$$

### E.6.4 Proof of Lemma E.4

Given any $\boldsymbol{\theta}'$,

$$\sum_{i=1}^d \left\| \bar{\phi}_{\boldsymbol{\theta}', i}(\boldsymbol{\theta}) \right\|_{\text{Lip}}^2$$

$$= \sum_{i=1}^d \left[ \sup_{\boldsymbol{\theta}_1 \neq \boldsymbol{\theta}_2} \frac{\left| \bar{\phi}_{\boldsymbol{\theta}', i}(\boldsymbol{\theta}_1) - \bar{\phi}_{\boldsymbol{\theta}', i}(\boldsymbol{\theta}_2) \right|}{\|\boldsymbol{\theta}_1 - \boldsymbol{\theta}_2\|_2} \right]^2$$

$$= \sum_{i=1}^d \sup_{\boldsymbol{\theta}_1 \neq \boldsymbol{\theta}_2} \frac{\left| \bar{\phi}_{\boldsymbol{\theta}', i}(\boldsymbol{\theta}_1) - \bar{\phi}_{\boldsymbol{\theta}', i}(\boldsymbol{\theta}_2) \right|^2}{\|\boldsymbol{\theta}_1 - \boldsymbol{\theta}_2\|_2^2}$$

$$\leq 2 \sum_{i=1}^d \sup_{\boldsymbol{\theta}_1 \neq \boldsymbol{\theta}_2} \frac{\left| \left( e^{-\|\boldsymbol{\theta}' - \boldsymbol{\theta}_1\|^2/\sigma} - e^{-\|\boldsymbol{\theta}' - \boldsymbol{\theta}_2\|^2/\sigma} \right) \frac{\partial}{\partial \theta_i'} V(\boldsymbol{\theta}') \right|^2}{\|\boldsymbol{\theta}_1 - \boldsymbol{\theta}_2\|_2^2}$$

$$+ 2 \sum_{i=1}^d \sup_{\boldsymbol{\theta}_1 \neq \boldsymbol{\theta}_2} \frac{\left| \frac{2}{\sigma} e^{-\|\boldsymbol{\theta}' - \boldsymbol{\theta}_1\|^2/\sigma} (\boldsymbol{\theta}_{1,i} - \theta_i') - \frac{2}{\sigma} e^{-\|\boldsymbol{\theta}' - \boldsymbol{\theta}_2\|^2/\sigma} (\boldsymbol{\theta}_{2,i} - \theta_i') \right|^2}{\|\boldsymbol{\theta}_1 - \boldsymbol{\theta}_2\|_2^2}.$$

For the first term on the right side of the inequality,

$$\sum_{i=1}^d \sup_{\boldsymbol{\theta}_1 \neq \boldsymbol{\theta}_2} \frac{\left| \left( e^{-\|\boldsymbol{\theta}' - \boldsymbol{\theta}_1\|^2/\sigma} - e^{-\|\boldsymbol{\theta}' - \boldsymbol{\theta}_2\|^2/\sigma} \right) \frac{\partial}{\partial \theta_i'} V(\boldsymbol{\theta}') \right|^2}{\|\boldsymbol{\theta}_1 - \boldsymbol{\theta}_2\|_2^2}$$

$$= \sum_{i=1}^d \left| \frac{\partial}{\partial \theta_i} V(\boldsymbol{\theta}') \right|^2 \sup_{\boldsymbol{\theta}_1 \neq \boldsymbol{\theta}_2} \frac{\left| \left( e^{-\|\boldsymbol{\theta}' - \boldsymbol{\theta}_1\|^2/\sigma} - e^{-\|\boldsymbol{\theta}' - \boldsymbol{\theta}_2\|^2/\sigma} \right) \right|^2}{\|\boldsymbol{\theta}_1 - \boldsymbol{\theta}_2\|_2^2}$$

$$= \|\nabla V(\boldsymbol{\theta}')\|^2 \left\| e^{-\|\cdot\|^2/\sigma} \right\|_{\text{Lip}}^2.$$

To bound the second term, by the symmetry of each coordinates, we have

$$\sum_{i=1}^d \sup_{\boldsymbol{\theta}_1 \neq \boldsymbol{\theta}_2} \frac{\left| \frac{2}{\sigma} e^{-\|\boldsymbol{\theta}' - \boldsymbol{\theta}_1\|^2/\sigma} (\boldsymbol{\theta}_{1,i} - \theta_i') - \frac{2}{\sigma} e^{-\|\boldsymbol{\theta}' - \boldsymbol{\theta}_2\|^2/\sigma} (\boldsymbol{\theta}_{1,i} - \theta_i') \right|^2}{\|\boldsymbol{\theta}_1 - \boldsymbol{\theta}_2\|_2^2}$$

$$= d \left\| \frac{2}{\sigma} e^{-\|\boldsymbol{\theta}\|^2/\sigma} \theta_1 \right\|_{\text{Lip}}^2.$$

This finishes the first part of the lemma.

$$\sum_{i=d}^{d} \left\| \bar{\phi}_{\boldsymbol{\theta}',i}(\boldsymbol{\theta}) \right\|_{\mathcal{L}_\infty}^2$$

$$= \sum_{i=d}^{d} \left\| e^{-\|\boldsymbol{\theta}'-\boldsymbol{\theta}\|^2/\sigma} \left( \frac{2}{\sigma}\boldsymbol{\theta}_i - \frac{2}{\sigma}\boldsymbol{\theta}_i' - \frac{\partial}{\partial \boldsymbol{\theta}_i'}V(\boldsymbol{\theta}') \right) \right\|_{\mathcal{L}_\infty}^2$$

$$\leq \sum_{i=d}^{d} 2 \left\| \frac{2}{\sigma} e^{-\|\boldsymbol{\theta}'-\boldsymbol{\theta}\|^2/\sigma} (\boldsymbol{\theta}_i - \boldsymbol{\theta}_i') \right\|_{\mathcal{L}_\infty}^2 + \sum_{i=d}^{d} 2 \left\| e^{-\|\boldsymbol{\theta}'-\boldsymbol{\theta}\|^2/\sigma} \frac{\partial}{\partial \boldsymbol{\theta}_i'}V(\boldsymbol{\theta}') \right\|_{\mathcal{L}_\infty}^2$$

$$\leq \sum_{i=d}^{d} 2 \left\| \frac{2}{\sigma} e^{-\|\boldsymbol{\theta}'-\boldsymbol{\theta}\|^2/\sigma} (\boldsymbol{\theta}_i - \boldsymbol{\theta}_i') \right\|_{\mathcal{L}_\infty}^2 + 2 \left\| e^{-\|\cdot\|^2/\sigma} \right\|_{\mathcal{L}_\infty}^2 \|\nabla V(\boldsymbol{\theta}')\|^2$$

$$\leq 2d \left\| \frac{2}{\sigma} e^{-\|\boldsymbol{\theta}\|^2/\sigma} \theta_1 \right\|_{\mathcal{L}_\infty}^2 + 2 \left\| e^{-\|\cdot\|^2/\sigma} \right\|_{\mathcal{L}_\infty}^2 \|\nabla V(\boldsymbol{\theta}')\|^2 .$$