[Reviews · NeurIPS 2020]

Review 1

Summary and Contributions: This paper extends Langevin dynamics (LD) MCMC with an additive self-repulsion term which encourages rapid exploration of the target distribution. The resulting Self-repulsive Langevin dynamics (SRLD) algorithm is interpreted as a single-chain version of Stein variational gradient descent. Theoretical results prove that the MCMC chain approaches the target distribution under some regularity assumptions. Empirical analysis shows realized gains over Langevin dynamics and Stein variational in terms of mixing to and exploration of the target

Strengths: The idea underlying SRLD is a simple one, implementation is straightforward, the paper is well-written, and a non-trivial theoretical analysis ensures correctness. The idea is a straightforward and reasonable extension of Langevin dynamics. Implementation of SRLD requires minimal additions to a Langevin dynamics sampler. The addition of a Stein repulsive term to SRLD requires only evaluation of the kernel and its gradient, and so adds little computational overhead. Empirical evaluation, while minimal, suggests improvements over LD and SVGD in synthetic and more realistic scenarios such as BNNs.

Weaknesses: The paper does not convincingly establish practical utility of SRLD. In particular, the empirical analysis shows modest (at best) improvements over LD and SVGD baselines. Evaluation of BNN inference on UCI datasets does not include more standard benchmarks such as Dropout MC (Gal and Ghahramani, 2017) or PBP (Hernandez-Lobato and Adams, 2015). It is unclear whether results reported in this paper are significantly different from those older works (note that Gal and Ghahramani achieve similar held-out-LL using a single-layer network). Another practical concern is that SRLB inference requires tuning a number of hyperparameters. In addition to the LD parameters (step-size and lag) SRLD includes kernel hyperparameters, thinning factor, and a weighting of the repulsive term. Moreover, the parameters are interdependent and ignoring dependence may violate algorithm correctness (remark @ L:245). There is discussion and guidelines in appendix A on hyperparameter tuning, but the paper doesn’t present any sensitivity analysis. A reasonable question to address is how sensitive inference is to repulsive term weighting. Practical utility aside, the theoretical analysis is correct only for strongly log-concave target distributions (Assumption 4.4). Corollary 4.1 does not list this assumption as necessary, but uses Thm. 4.3, which does, and so it should be included.

Correctness: I did not find any significant technical issues.

Clarity: The paper is well-written and clear.

Relation to Prior Work: Prior work is clearly discussed.

Reproducibility: Yes

Additional Feedback: POST REBUTTAL UPDATE: I have considered the author's response and will keep my original score. L:315 references A.5 which doesn’t exist The extension to general dynamics in Sec. 5 is speculative, not validated in any way, and is not sufficiently formulated (e.g. B_t in the first equation is never defined) L:100 references wrong equation (1) Presentation of SRLD assumes support over all of R^d. The authors should discuss Broader impacts discussion is limited


Review 2

Summary and Contributions: The author proposes a self-repulsive dynamics that combines Langevin dynamics and Stein variational gradient descent (SVGD) called Self repulsive Langevin dynamics (SRLD). The author also analysed the proposed method theoretically in terms of its stationary distribution, mean-field and continuous-time limit. The advantage of the proposed method is that the repulsive force enables the SRLD to be far away from the past samples and visits the previously unexplored regions. The ESS and auto-correlation within the SRLD chain are improved compared to just Langevin dynamics. ------------------------ After reading the response, the author promised to include more comparisons. However the reason I want to see more comparisons is that I am not fully sure on the effectiveness of the repulsive force in very high dimensions (not the repulsive force in the paper, but the second term in g). In the author's response, they claim the vanishing repulsive force can be addressed by incresing \alpha, which is not fully correct. Because increasing \alpha would also increase the confining term as well, thus, overall, the actual repulsive force (2nd term in g) is still small. I suspect the repulsive force can be even negligible in high dimensions. Thus, I would love to see comparisons with advanced SGMCMC method on high dimensional problems to verify the usefulness of the repulsive force.

Strengths: The intuition of the proposed method is easy to understand, it is simply an addition of Langevin dynamics and SVGD. The argument about its advantages is reasonable. Theoretically, the author proved that the stationary distribution of the proposed method is indeed the target distribution in the limit of infinite past samples and infinitesimal step size.

Weaknesses: The presentation of the method can be improved a lot. For instance, I did not find the related work section, instead, the relevant literature is scattered throughout the main text. I think the author should group them in the related work section. I also did not find the conclusion section. I also did not understand some parts of the proof and I will elaborate in the following.

Correctness: I have looked into the theorem about the stationary distribution and a quick scan over the rest. They seem to be correct, but I am not fully sure.

Clarity: The presentation of the method is easy to understand but the rest of the paper is not very well-written, especially the structure of the paper.

Relation to Prior Work: The author mentioned the proposed method is different from the previous work that tries to combine SVGD with Langevin dynamics in terms of motivation and implementation. But, I think adding a related work section would make the reader easier to appreciate the novelty of the SRLD and its differences compared to the previous method. Also, the section number of the Appendix is wrong.

Reproducibility: Yes

Additional Feedback: 1. Adding related work section. For example, relevant literature about SGMCMC, SVGD, the method to combine SVGD and Langevin dynamics. 2. Consider adding a conclusion section? 3. In line 305, where is Appendix A.4? In line 315, where is A.5? 4. In Appendix, why contextual bandit section is a subsection of BNN on UCI dataset? 5. Have you considered adding some advanced SGMCMC baselines? Like preconditioned SGLD? 6. Does your method have performance advantages compared to parallel SGMCMC? 7. As the proposed method is self repulsive, thus, when sampling from multimodal distribution, does it avoid being trapped in a local mode? Maybe consider a multimodal distribution in the synthetic experiment. 8. In equation 1, the Stein Repulsive term involves both repulsive and confining terms in SVGD. So this confining term in SVGD also provides a drift force. So maybe Stein Repulsive is not a good name for this term? 9. It is well-known that the SVGD has a vanishing repulsive force problem, especially in high dimensions. This will make the repulsive force too weak to repulse any particles. Have you noticed this problem in your experiments? One good indicator is to examine the ratio between the confining terms and repulsive terms in SVGD like in Zhuo J, 2018. 10. I am a bit confused about how the author obtained the equation about the density evolution equation (The one below line 224 and also equation 5). From my understanding, these are the direct application of the Fokker-Planck equation. However, the Fokker-Planck equation is only for Ito process, where the drift term can only be relevant to the current state \theta_t and time t. But in the author's formulation, the drift term contains the past density \bar\rho_t^M? My question: can you still use Fokker-Planck equation? Please elaborate more on this. 11. In line 205, should it be \nabla V(\theta)? 12. In equation 4, should it be \theta_t instead of \theta_k? References: Zhuo J, Liu C, Shi J, et al. Message passing stein variational gradient descent[C]//International Conference on Machine Learning. 2018: 6018-6027.


Review 3

Summary and Contributions: This paper proposes Stein self-repulsive dynamics, which leverages Stein variational gradient as a repulsive force to push the new samples from Langevin dynamics (LD) away from the samples in its past trajectory. This can encourage and improve diversity of the LD samples for nay intractable un-normalized distributions. The authors also show the theoretical analysis that the asymptotic stationary distribution remains correct even with the addition of the repulsive force, thanks to the special properties of the Stein variational gradient. The experiments are conducted on 2D toy distributions, several small datasets for BNNs and Contextual Bandits. The proposed SRLD shows better performance than two baseline methods, including SVGD and LD.

Strengths: S1: The idea is interesting: incorporating the historical information/samples, and use them to drive the new samples away using Stein variational gradient. The technique can be easily used as plug-in term for any LD algorithms. S2: The presentation is good. In particular, Figure 1,2,3 clearly shows the advantages of the proposed Stein self-repulsive term in LD. S3: The proof on the asymptotic convergence to stationary distribution seems non-trivial, it can be a technique strength of this submission. PS: I am not an expert in proving this, and can not judge its value.

Weaknesses: While the idea itself is worthwhile, the empirical evaluation of the proposed technique is weak. W1: It is only compared the most basic SG-MCMC (ie, SGLD) and SVGD variant. I believe the proposed Stein self-repulsive term is a general technique for any SG-MCMC (Yes, this is indeed an advantage), it will help improve the paper (showing its generality and position in the literature), if the authors can compare and further combine with state-of-the-art variants of SG-MCMC[1,2,3], and see how it performs: whether it is comparable with them, and even outperform them when combined. Further, there are other sampling methods (e.g., based on generative adversarial methods [4]) for un-normalized distributions. How is the proposed method compare with them? [1] Stochastic Gradient Hamiltonian Monte Carlo, ICML 2014 [2] Preconditioned stochastic gradient Langevin dynamics for deep neural networks, AAAI 2016 [3] Cyclical Stochastic Gradient MCMC for Bayesian Deep Learning, ICLR 2020 [4] Adversarial Learning of a Sampler Based on an Unnormalized Distribution, AISTATS 2019 W2: The datasets/tasks used in this paper is not really convincing. For example, the UCI datasets are the most comprehensive datasets used in this paper. However, the performance variance on these datasets are large, even with multiple runs. There are dozens of papers reporting similar results on these datasets. It is easy to find several papers that outperform the proposed methods. My point is that the authors can consider larger models/datasets for evaluation. For example, please consider the experiment settings in [3], with results on ResNets on CIFAR (even on ImageNet if possible). It is interesting to effectively explore the multi-mode distributions of large BNNs to see if it can yield further improvement.

Correctness: The algorithm is correct.

Clarity: yes

Relation to Prior Work: Please see my concerns in W1

Reproducibility: Yes

Additional Feedback: Thanks for the rebuttal, my concern remains, especially on improving the significance of the proposed algorithms with SoTA experiments/settings.

[Author Response · NeurIPS 2020]

We thank all the reviewers for their time and valuable feedback. We will improve the draft based on your comments, and we hope you could raise your evaluation if we address your concerns.

**Reviewer 2: (On benchmarks)** We agree that including these baseline can be useful for the potential readers and we will add them in next version. We also want to notify the reviewers that it has been well tested in [2] that SVGD outperforms PBP so our baselines are actually stronger than PBP.

**(On hyper-parameter tuning)** We didn't tune the kernel bandwidth but simply apply the widely used median trick [2]. The thinning factor is common in MCMC and has been used for decades. We will add sensitive analysis on that. Our ablation study on the repulsive term weighting ($\alpha$) is included in Appendix B.2 (we will give pointer on that in next version). Also, since that we propose a specific way to tune the $\alpha$ (as shown in appendix A) and its usefulness has been demonstrated as in all the experiment we use this criterion to select $\alpha$, we believe the sensitive analysis of $\alpha$ is sufficient and the proposed criterion for selecting $\alpha$ is reliable in practice.

**(On the Presentation)** Sorry for the appendix indexing issues. Appendix A.5 is Appendix C.1. $\mathcal{B}_t$ denotes the Brownian motion and we will add definition. Our section 5 demonstrates that our Stein repulsive gradient can be applied to general dynamics as the limiting system is able to produce correct targeted distribution. As the later discretization and large particle approximation is almost identical to that in SRLD, we omit the details. We will give more discussion on that. The reference in L.100 should be referred to equation between L.80 and L.81 and we will fix it as well as the other clarity issue you mentioned.

**Reviewer 3: (1, 2)** We will add a related work section that collect the literature review in the current version, as well as a conclusion section in the next version.

**(3, 4)** Sorry for the index problem. Appendix A.4 should be Appendix C and Appendix A.5 should be Appendix C.1. We will give contextual bandit an individual section in the next version.

**(5, 6)** Notice that our repulsive term can be applied to any SG-MCMC and thus we believe the current experiment using the simple Langevin dynamics is able to show the usefulness of the repulsive term. We will test some more advanced SG-MCMC method.

**(7)** We have a synthetic experiment on sampling multi-mode distribution (see Appendix B.1 and B.3).

**(8)** We agree that the name repulsive is a little bit not exact, but given that this term provides repulsive force in practice, we believe this name is intuitive.

**(9)** Our method allows us to adjust the $\alpha$ to increase the magnitude of repulsive gradient compared with Langevin term. Please see line 440-443 for more details. The issue of kernel method in high dimension is a problem independent with this work and existing technique to mitigate this issue (e.g. [1]) can also be applied to our method.

**(10)** The density evolution is not derived by the direct application of standard Fokker-Planck equation. The derivation is standard and mostly calculation (similar to the one in Appendix A.3 of [5]) and thus we omit it previously. We will give details on the derivation in the next version.

**(11, 12)** We don't need that strong condition and the current statement is correct. And yes, it is $\theta_t$ in equ (4), sorry for the typo.

**Reviewer 4: (W1)** Thanks for the point! We agree it would be more complicate to conduct experiment on different advanced MCMC and we will add them in the next version. Due to the time limit, we are unable to give result on that currently. However, we also believe the current experiment design is sufficient and good. We believe the main focus on the experiment is to demonstrate the usefulness of the proposed Stein repulsive gradient and thus it is the best choice to work on the simple Langevin dynamics, which has the least hyper-parameters to tune.

**(W2)** We agree that the experiment on UCI has large variance and thus we conduct statistical testing (based on matched pair t-test, see, e.g. caption of Table 1) for all the results we report. As most result are statistically significant, we believe our result on UCI dataset is convincing. Besides, as we use different setting and data split for UCI dataset, the results in other papers are not directly comparable to ours.

We agree that it might be interesting to test the behavior of our method on large BNN model. However, due to the computation resource constraint, we are not able to maintain, e.g., 20 checkpoints of large neural nets at the same time and thus we currently are not able to do such experiment. Beside, we think the main focus of [3] and ours is different, which makes their experiment design not suitable to us. [3] aims to use Bayesian-like ensemble to improve the prediction accuracy rather than approximate the posterior (thus they only use very small number of particles that cannot approximate the posterior). Our aim is still on a better MCMC algorithm that has better posterior sample quality.

[1] Understanding and accelerating particle-based variational inference.
[2] Stein variational gradient descent: A general purpose bayesian inference algorithm.
[3] Cyclical Stochastic Gradient MCMC for Bayesian Deep Learning.
[4] Dropout as a bayesian approximation: Representing model uncertainty in deep learning.
[5] Stein Variational Gradient Descent as Gradient Flow.


[Meta-Review · NeurIPS 2020]

This paper extends Langevin dynamics (LD) MCMC with an additive self-repulsion term which encourages rapid exploration of the target distribution. Strengths: - the idea is simple to implement and the paper is well-written - non-trivial and elegant theoretical analysis proves correctness Weaknesses: - empirically the method does not very strongly outperform Langevin dynamics - it is unclear how the method works in high dimensions